# Changes over the Last 35 Years in Alaska's Glaciated Landscape: A Novel Deep Learning Approach to Mapping Glaciers at Fine Temporal Granularity

Ben M. Roberts-Pierel [1,*], Peter B. Kirchner [2,3] , John B. Kilbride [1] and Robert E. Kennedy [1]

1 Geography Program, College of Earth, Ocean and Atmospheric Sciences, Oregon State University, Corvallis, OR 97331, USA
2 Southwest Alaska Monitoring Network, National Park Service, Anchorage, AK 99501, USA
3 Department of Ecosystem and Conservation Science Franke College of Forestry and Conservation, University of Montana Forestry, Missoula, MT 59812, USA
* Correspondence: robertsb@oregonstate.edu

**Abstract:** Glaciers are important sentinels of a changing climate, crucial components of the global cryosphere and integral to their local landscapes. However, many of the commonly used methods for mapping glacier change are labor-intensive and limit the temporal and spatial scope of existing research. This study addresses some of the limitations of prior approaches by developing a novel deep-learning-based method called GlacierCoverNet. GlacierCoverNet is a deep neural network that relies on an extensive, purpose-built training dataset. Using this model, we created a record of over three decades long at a fine temporal cadence (every two years) for the state of Alaska. We conducted a robust error analysis of this dataset and then used the dataset to characterize changes in debris-free glaciers and supraglacial debris over the last ~35 years. We found that our deep learning model could produce maps comparable to existing approaches in the capture of areal extent, but without manual editing required. The model captured the area covered with glaciers that was ~97% of the Randolph Glacier Inventory 6.0 with ~6% and ~9% omission and commission rates in the southern portion of Alaska, respectively. The overall model area capture was lower and omission and commission rates were significantly higher in the northern Brooks Range. Overall, the glacier-covered area retreated by 8425 km$^2$ (−13%) between 1985 and 2020, and supraglacial debris expanded by 2799 km$^2$ (64%) during the same period across the state of Alaska.

**Keywords:** glacier change; glacier inventory; deep learning; neural network; remote sensing; Landsat

## 1. Introduction

Glaciers are crucial components of the global cryosphere and iconic symbols of the landscapes they occupy and shape. Meltwater from glaciers constitutes an important component of freshwater discharge through river systems in many regions of the world, impacting water availability, temperature, sediment, and nutrient conditions in their associated stream, river, and ocean ecosystems [1–4]. Glaciers affect the Earth's energy balance, and glacier discharge is an important contributor to global sea level rise [5–7]. With ~60,000 km$^2$ of glacier-covered area, the state of Alaska is one of the most heavily glaciated regions in the world outside the polar regions. Alaska encompasses many different glacier types, from very small (<0.1 km$^2$) mountain glaciers to North America's largest and longest valley, tidewater, and piedmont glaciers. These systems contribute ~75% of the freshwater input and significant nutrient flux to the Gulf of Alaska, supporting one of the world's most productive fisheries [8].

The effects of climate change are being felt more immediately in the Arctic than in other regions of the world [9–11]. As a result, glacier-covered areas in Alaska are being adversely affected and most debris-free ice is receding and thinning [6,12,13]. However,

the trends for debris-covered glaciers and tidewater glaciers are more complex, with expansions of supraglacial debris and mixed impacts on area and volume changes [14–18]. Cataloging ongoing changes and improving sources of uncertainty in understanding changes in Alaska's glaciated landscape are crucial for many reasons, including land and water management decisions, economic activity, and cultural purposes. Due to the importance of Alaska's glaciers for the state, as well as the global cryosphere, it is necessary to understand their trajectory and magnitude of change by mapping these trends efficiently with fidelity and regularity.

Previous studies have cataloged the overall spatial extent of Alaska's glaciers at various points in time [19–22], while others have investigated additional glacier parameters such as mass balance [6,12] and debris cover characteristics [14,17,23]. The existing literature contributes to a solid understanding of the broad scope of glacier-covered area in Alaska, but also leaves substantive gaps that diminish its utility for rigorously evaluating trends.

The first limitation in existing research arises from the classification techniques used to map glacier areas. Most existing products were created with automated classification followed by extensive manual editing (semi-automated). Semi-automated approaches build from high to moderate spatial resolution (~2–30 m) spaceborne optical remote sensing data, leveraging ratios between spectral bands or using a pixel-based classification approach. Manual editing helps as a post hoc mapping technique, but is labor-intensive, time-consuming, and subjective. At best, manual techniques are feasible for a broad spatial scope or broad temporal scope, but not both. Exceptions to these issues exist [17,23–27], but even those studies suffer from two additional limitations.

The second shortcoming of existing work is that it is limited in either temporal or spatial scope. Even the studies that used a broad time range typically missed temporal granularity or covered a very small area. Most studies which focused on long-term temporal change took an *end-point approach*: an early date range composite was compared with a contemporary date range composite to capture a shift over time [19,27]. End-point composites miss important fluctuations in the intervening decades. Where approaches with denser time-series resolution have been used, they are often for short time periods, limited geographic scope and are pixel-based [28,29]. Pixel-based approaches lack the spatial context of object-based approaches, which can be particularly problematic for mapping a highly variable target such as glaciers. Where object-based approaches have been used to map debris-covered glaciers, they are limited by small geographic areas [27] and sometimes short study periods [26]. An important exception is a global time series of glacier change developed by Hugonnet et al. [6]. Although useful, that work was constrained to a starting year of 2000 and the mapped change did not explicitly consider debris-covered area as a separate class.

The third limitation of existing work is a focus on a single glacial parameter. For example, Hugonnet et al. (2021) focused only on mass loss, with no distinction between debris-free glaciers and supraglacial debris. Others [14,17] have produced datasets of global scope, but focused exclusively on supraglacial debris for a limited contemporary time window. Likewise, McNabb and Hock [30] included a long time period (1948–2012) but focused exclusively on capturing changes in Alaska's tidewater glaciers. Without considering multiple glacier types at a fine cadence over a broad area, spatial and temporal interactions or trends occurring simultaneously in different types of glaciers may be missed.

Our study addresses these limitations by leveraging the rapidly evolving methods of deep learning to construct a +35-year time series at two-year intervals for all glacier-covered areas in the state of Alaska. We used an image processing and classification model framework to capture overall glacier-covered area and to distinguish between debris-covered glacier (supraglacial debris) and debris-free ice. The resultant dataset offers higher temporal granularity than any existing product over a longer time period (two-year composites for 1985–2020), and it does not employ the manual editing of glacier outlines in the post processing steps. Although focused on the state of Alaska, we expect that these

methods and associated findings have broader implications for identifying changes in global glacier-covered area.

Our approach includes several important methodological advances. Firstly, we incorporate recent research from the computer vision subfield of deep learning techniques to improve the accuracy of our glacial mapping products [31,32]. Secondly, our neural network produces maps of glacier extent by semantically segmenting the imagery (i.e., each pixel of the input image is labeled simultaneously), producing highly accurate glacier maps with well-defined classification boundaries. In contrast, existing studies have classified glacier extent using pixelwise, non-parametric classification techniques (e.g., Random Forests, Support Vector Machines) [24,33,34]. Finally, instead of producing a single classification, we develop a multi-decadal time series of maps which document the evolution of glacier extent across multiple classes simultaneously.

To introduce and describe this novel and globally applicable method for mapping glacier and glacial debris-covered areas we: (i) present a publicly available dataset of glacier and glacial debris-covered areas in Alaska, with a robust classification of uncertainty; (ii) characterize the location, magnitude, and trends of changes in debris-free ice and supraglacial debris glacier-covered areas; and (iii) characterize the relationship between modeled temperature gradients and historic changes in glacier-covered area in Alaska.

*Study Area*

The state of Alaska has approximately 60,000 km$^2$ of glacier-covered area, with a wide range of mean annual temperatures, terrain, and glacier types. Elevations across the state range from 0 to 6190 m ASL [35]. The region is bounded on the north and east by the seasonally frozen Beaufort, Chukchi and Bearing seas, and on the south by the deeper, warmer waters of the North Pacific's Gulf of Alaska. Approximately 99% of Alaska's glacier-covered area is found in the mountainous terrain of the southern region, within approximately 200 km the Gulf of Alaska (Figure 1). The remaining 1% is found in the Brooks Range in the northern area of the state (Figure 1). Although these two regions have some similarities, there are many differences in glacier characteristics. The southern region, adjacent to the Gulf of Alaska, has numerous tidewater, valley, and piedmont glaciers and icefields, in addition to many cirque and alpine glaciers. Conversely, the northern region, centered on the Brooks Range, is composed of smaller, shaded cirque and alpine glaciers at a higher latitude, requiring different approaches from optical remote sensing. To ensure the continuity of predictor variables and outputs, we defined our study area exclusively as the state of Alaska. This choice was primarily due to a dearth of current, spatially continuous, high-resolution digital elevation models outside the state boundary.

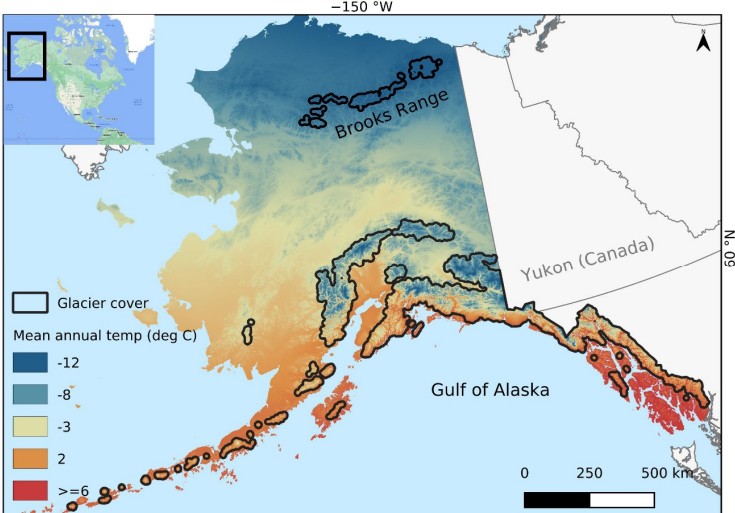

**Figure 1.** Climate normal (1981–2010) calculated from mean annual temperature from Daymet v4 [36], including simplified outlines of glacier-covered areas within Alaska.

## 2. Materials and Methods

Our workflow consisted of a series of algorithms that created a spectrally homogenous time series of optical imagery, followed by an initial classification of debris-free glacier ice, and finally, the creation of multi-class maps of glacier cover change over the study time period (1985–2020) using a deep neural network (Figure 2: GlacierCoverNet Training). We created the initial optical imagery time series from the Landsat archive using the LandTrendr algorithm [37] which was run on the Google Earth Engine (GEE) platform [38,39]. To train the deep learning model, we built a training dataset composed of predictor variables from LandTrendr outputs, topographic information, and a class label.

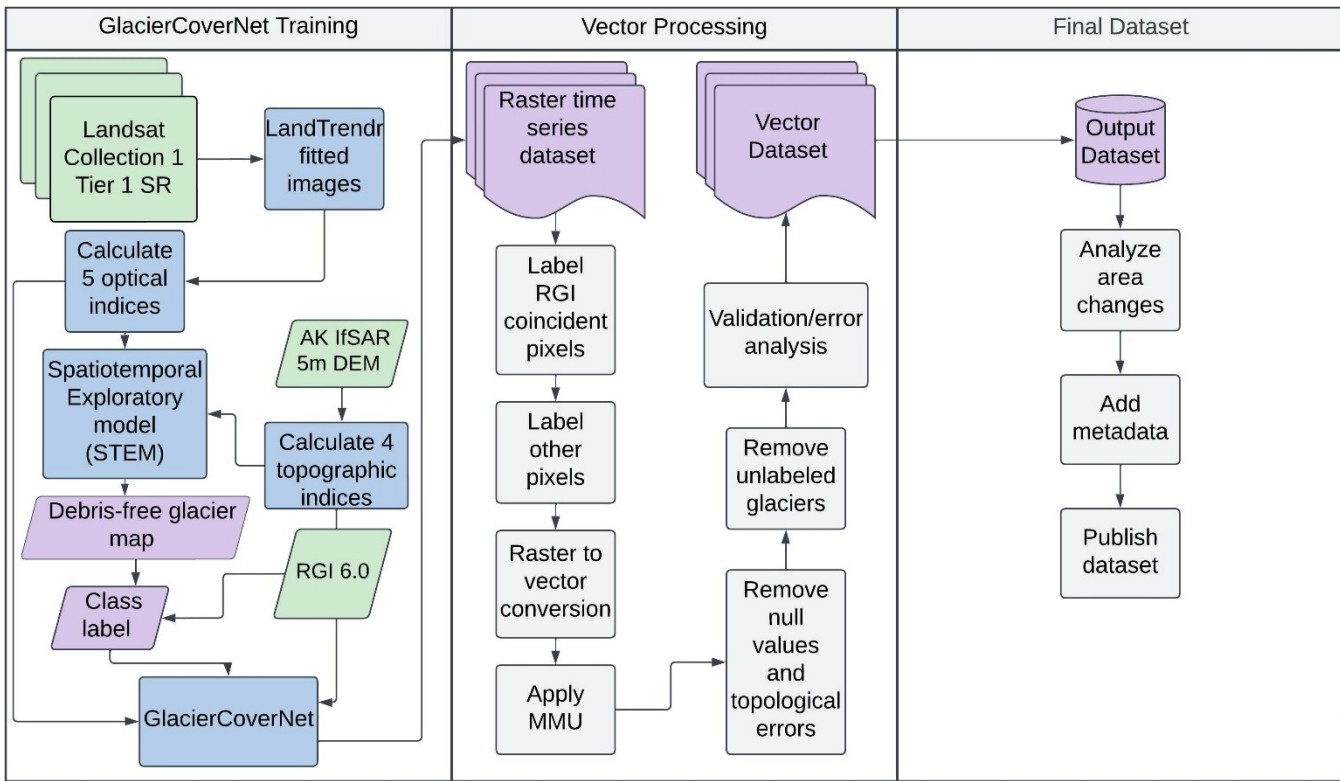

**Figure 2.** Conceptual workflow showing data inputs in green, algorithms in blue, computational outputs in purple, and post hoc processing steps in gray.

The class label was created by first identifying debris-free glacier with a spatiotemporal exploratory model (STEM) approach [40]. We then combined this classification with the RGI 6.0 boundaries, assuming that the difference in area was supraglacial debris (Figure 2: GlacierCoverNet Training).

We then used a set of ~650,000 nine-channel predictor variable and class label image chips to train a deep neural network, hereafter referred to as GlacierCoverNet. The object-based classification approach, coupled with the large training dataset, produced maps of debris-free glaciers, supraglacial debris, and overall glacier-covered area with high spatial fidelity across most of the study domain without manual editing of glacier outlines.

To prepare the dataset for public use, we carried out a series of post hoc editing steps to convert from default raster outputs to vectors, assigned RGI 6.0 labels, and conducted an error analysis (Figure 2: Vector Processing). We then added final metadata for archival and public dissemination (Figure 2: Final Dataset).

### 2.1. Landsat Archive and LandTrendr

LandTrendr (LT) is a temporal segmentation algorithm designed for disturbance detection that also builds a radiometrically and temporally consistent time series of optical imagery, making it well suited for generating the imagery required for this project [37,40–42].

The LT algorithm linearly segments the time series of spectral values and adjusts spectral values based on those segments, thus removing ephemeral noise that can be introduced by atmospheric haze, clouds, smoke, variations in image acquisition date, and sensor parameters [43,44]. The LT model was run on a time series of annual medoid composites [45] of Landsat imagery. Our compositing window spanned from 20 July to 20 September to obtain a sufficiently wide time window to maximize the number of Landsat overpasses while minimizing conflation of glacier ice with out of season snow.

We utilized all available 30 m resolution images from the Landsat program from 1985 to 2020 collected over Alaska, resulting in biannual composites (see below) that ended on the most recent year of imagery when work was conducted (2020). We used images from Landsat 4 TM, Landsat 5 TM, Landsat 7 ETM+, and Landsat 8 OLI, employing coefficients outlined by Roy et al. [46] to bridge the change in radiometric properties between the pre-Landsat 8 and Landsat 8 mission sensors. We rely here on the LT algorithm to fill in missing data and artifacts from the known SLC off error that resulted from the failure of the scanline corrector on the Landsat 7 ETM+ instrument. We only used the data included in the Collection 1 Tier 1 surface reflectance archive for this project.

Some of our domain, particularly near the Gulf of Alaska, was frequently obscured by clouds. To address that challenge, we created two-year composites of cloud-masked imagery, increasing the number of possible input images relative to a one-year cadence. Therefore, in the processed images with the label year 2020, input data included images from 2019 and 2020. For cloud masks, we started with the standard Landsat CFmask [47] (including cloud and shadow), then augmented using normalized difference snow index (NDSI) and shortwave infrared masks to reduce over-masking by the CFmask in glacier-covered and commonly snow-covered regions where high reflectance was sometimes mistaken for clouds. After cloud masking, areas of missing data were filled using LT to interpolate between two-year composites. After comparing different composite lengths with the availability of pixels not removed by the masking process, we determined that two-year composites provided sufficient data to build image composites while maintaining high temporal granularity. The outputs of this process were 18 biannual image composites with full spatial coverage over the state, including areas with high seasonal cloud cover.

### 2.2. Predictor Variables

Both the STEM and GlacierCoverNet models utilized a ten-band raster that incorporated five predictors from the LT-interpolated Landsat composites, four predictors derived from the Alaska 5m IfSAR elevation product [48] and a class label (Table 1). General predictor variable selection aimed to balance meaningful spatial and spectral inputs with computational efficiency.

These optical predictors maximized the depth of spectral information derived from the Landsat imagery by incorporating indices aimed at capturing different components of heterogeneous landcover. In addition to the shortwave infrared (SWIR), green, and red bands used to derive NDSI and used for band ratioing [28,29,54], we used NDVI and NBR to assist in better delineation of bare ground and vegetation in the proglacial environment [55]. Linear transformations of the Tasseled cap indices have also been used for a wide range of relevant remote sensing applications e.g., [56,57]. Here, they improved the delineation of debris-free glacier, supraglacial/proglacial debris, and proglacial vegetation. Although the thermal bands have been used successfully in the past to map supraglacial debris, we opted not to use them here, mostly because of the difference in spatial resolution and lack of time series consistency.

**Table 1.** Summary of optical (1–5) and topographic (6–9) predictors used for the spatiotemporal exploratory model (STEM) and GlacierCoverNet models.

| Band Number | Band Name | Derivation Source |
|:---:|:---|:---:|
| 1 | Normalized difference snow index (NDSI) | [49] |
| 2 | Normalized difference vegetation index (NDVI) | |
| 3 | Normalized burn ratio (NBR) | [50] |
| 4 | Tasseled Cap Brightness | [51] (p. 13) |
| 5 | Tasseled Cap Wetness | [51] (p. 13) |
| 6 | Curvature | [52] |
| 7 | Aspect intensity (north) | [53] |
| 8 | Aspect intensity (south) | [53] |
| 9 | DEM | [48] |
| 10 | Class label | Purpose built |

The topographic indices derived from the IfSAR elevation dataset are robust for identifying features related to snow deposition and glacier cover [53]. Their inclusion augmented indices derived from optical data by aiding in the identification of glacier surfaces where conflation with snow cover was common. The indices of curvature and aspect intensity include slope and aspect in their derivation, providing information on the shape, surface roughness, and landscape positions of glaciers.

*2.3. Glacier Semantic Segmentation Dataset*

We used an encoder–decoder neural network architecture (GlacierCoverNet) to semantically segment the multispectral satellite imagery and topographic predictors. Semantic segmentation differs from traditional pixel-based classification approaches such as random forest [58] or support vector machines [59], which work by taking a vector of features summarizing pixel-level attributes and mapping them to a set of outputs. Instead, the GlacierCoverNet model used a 128 × 128 ten-channel image as input and output a classified 128 × 128 map. The value of each pixel in the final image was assigned to one of the label classes: no glacier, supraglacial debris, debris-free glacier.

2.3.1. Class Label Generation

A deep learning model requires a target output, hereafter referred to as the class label. To create the class label raster, we combined the outputs of the STEM model (discussed below) with the RGI 6.0 boundaries, assuming the areas between the STEM classification (debris-free ice) and the RGI boundary were debris-covered glacier areas. This differencing approach follows other studies that have used it to approximate debris-covered glacier area [17]. Although there is uncertainty associated with this step, deep learning models have been shown to generalize well even in the presence of significant noise in the class label layer [60–64].

The STEM was used to build a contemporary landcover map with a focus on debris-free glacier ice. The STEM model was first used to extend the temporal coverage of the Landsat-derived National Landcover Database (NLCD) [65]. Hooper and Kennedy [40] used these finite NLCD maps (2001 and 2016) to define map classes and constrain the sampling of predictor variables. Here, we substituted a glacier probability layer for the NLCD data to identify the likelihood of glacier presence or absence. The glacier probability layer was constructed as a stack composed of data from the 2016 National Landcover Database (NLCD), the Randolph Glacier Inventory 6.0 (RGI 6.0) [20,22], and the NASA MEASURES Inter-mission Time Series of Land Ice Velocity and Elevation (ITS-LIVE) [66,67] datasets. These datasets were combined to produce a simple 0–3 scale, where a pixel with the value 0 had none of the datasets identifying glacier-covered area, whereas pixels with the value 3 were labeled glacier in all the datasets. The utility of the probability dataset was twofold: first, it provided the STEM model with multiple glacier-specific strata to classify

because the original target (NLCD) is not focused on the cryosphere. This multi-class target layer was also used because there was no ancillary training/reference dataset that combined the overall glacier-covered area and supraglacial debris for our domain. If such a dataset existed, it would have been a logical choice for a target layer. Second, the probability dataset evaluated the performance of the pixel-based classification approach where there was no prior information on debris-covered glacier. When using STEM outputs in the class label, values of two and above were included because there was too much noise in the value 1 category to be of use in the model.

The predictor variables for the STEM model were spectral indices calculated from the LT outputs and topographic indices generated from the Alaska 5 m IfSAR DEM (Section 2.2). The STEM model used a group of regression trees which were deployed inside a series of randomly shuffled, spatially overlapping support sets, essentially running the classification $n$ times for every pixel in the domain [40,68]. These cumulative results were then condensed by taking the mode value on a pixel-wise basis to produce the classified output.

The output of the STEM model was a binary (glacier/no glacier) raster based on 2008–2010 optical imagery. This year range was chosen because it incorporated the median start year for the glacier outlines in RGI 6.0 for Alaska. We found using the STEM model to be effective for mapping debris-free ice and comparable to the widely used band ratioing approach [28,69–71]. We combined the STEM output with RGI 6.0 to define debris-covered glacier and created a three-class label raster (no glacier, debris-covered glacier, and debris-free glacier). To prepare this class label raster for use in the GlacierCoverNet model, we conducted limited manual editing to remove isolated artifacts from masked pixels in the input images and then employed a nearest neighbor analysis (GDAL sieve) to remove speckling left by the pixel-based STEM classification.

### 2.3.2. Training Data Generation

To build the training dataset for the GlacierCoverNet model, we gridded our class label raster using 256 × 256 pixel partitions. After testing 128 × 128, 256 × 256, and 512 × 512 partition sizes, we selected 256 × 256 to balance a meaningful spatial unit (i.e., a unit of analysis that captures informative spatial features such as a glacial lobe) with computational performance. The 256 × 256 partitions were randomly split with 80% devoted to training and 20% to testing (Figure 3). Training partitions were then systematically subsampled into 128 × 128 pixel image chips ($n = 823,976$) with each chip comprising our nine predictor variable channels and the class label. A stride of 32 pixels was used after testing multiple versions. This selection was to balance dataset size (smaller stride) with dataset variability (larger stride). To prevent a temporal mismatch between the predictors and the labels, the spectral predictors were extracted from the biannual composite closest to the average date of the RGI perimeters contained in the spatial partition. These dates were identified by the BgnDate attribute field included with the RGI 6.0. A fundamental weakness of the RGI 6.0 is tracking of temporal trends. A subset of the semantic segmentation dataset was allocated to model training (80%, $n = 655,452$). The remaining data were assigned to a validation set (20%, $n = 143,519$), which was used to provide an estimate of the deep learning algorithm's performance.

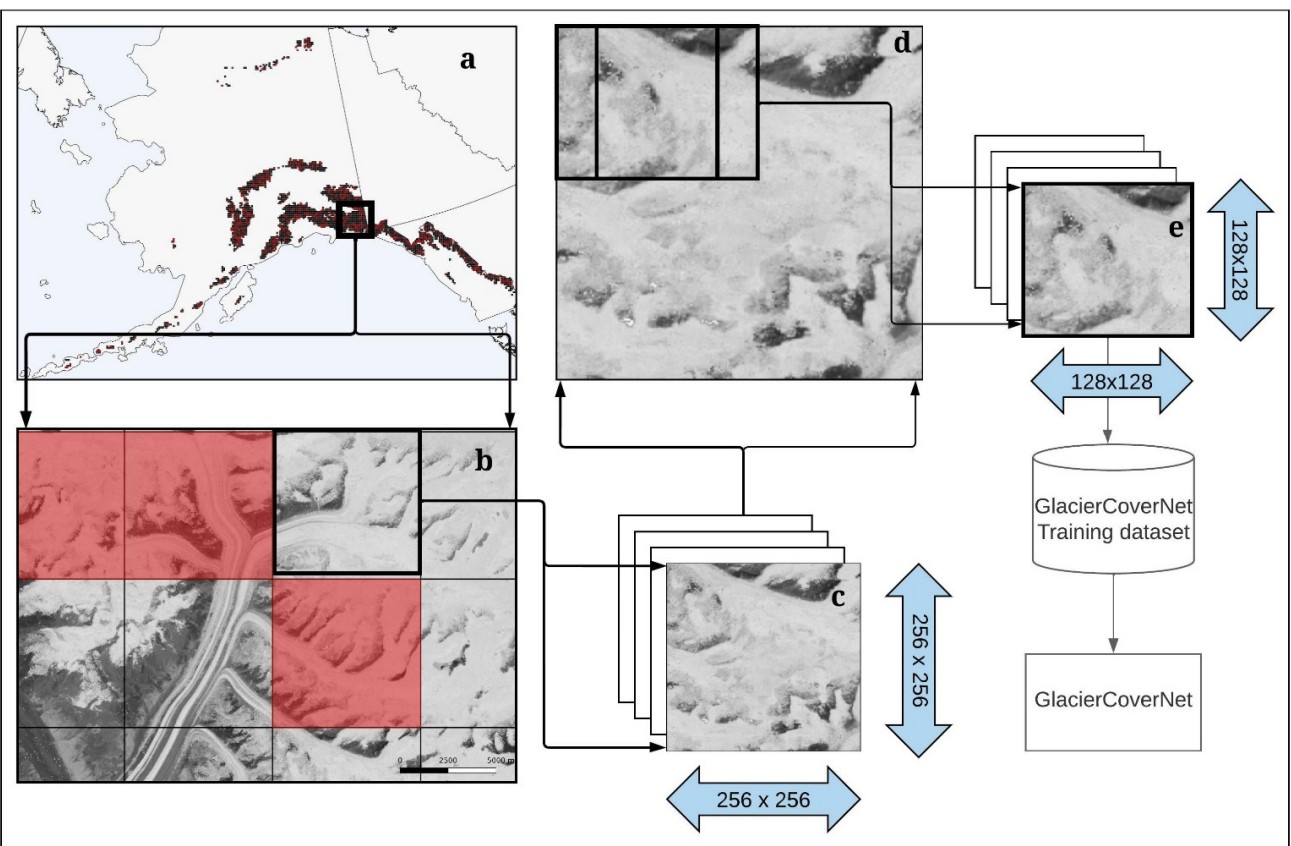

**Figure 3.** Training data generation. (**a**) Study area partitioned into 20% testing (red) and 80% training (black), (**b**) zoom of partitioned area with testing (red) and training (black outline), (**c**) 10-band 256 × 256 image partition, (**d**) moving window sampling process, and (**e**) final 128 × 128 image chips used for GlacierCoverNet model training and testing. Note that this is a conceptual diagram and is therefore not to scale.

### *2.4. GlacierCoverNet Architecture*

Encoder–Decoder Structure

The GlacierCoverNet model is an encoder–decoder neural network which uses a ResNeSt-101 convolution neural network and a Pyramidal Scene Parsing Network (PSP-Net) module to semantically segment glaciers and glacier debris using a combination of spectral and topographic inputs [72,73]. ResNeSt-101 is a deep convolutional neural network which modifies the ResNet101 architecture with split-attention blocks (i.e., a group of convolutional layers, activation functions, and transformations) to improve the network's accuracy and efficiency (Appendix A.1). The split-attention blocks incorporate residual connections, multiple pathways in each block, and channel-wise soft attention, which have empirically been shown to improve the accuracy of models on computer vision tasks [74–76]. Gaussian Linear Error Units were used as the non-linear activation functions for both the ResNeSt-101 encoder and the PSPNet decoder [77]. The PSPNet decoder combines information encoded at multiple resolutions by the ResNeSt-101 encoder to produce a set of information-dense feature maps. This study did not explicitly use techniques such as L1 or L2 regularization. Finally, the feature maps produced by the PSPNet were up-sampled to match the resolution of the input imagery, enabling a classification label to be assigned to each pixel in the input satellite imagery (Appendix A.1).

The spatial morphologies of glaciers in northern Alaska (e.g., Brooks Range) are largely distinct from many glaciers in southern Alaska, which necessitated a two-phase model development strategy. The majority of the training dataset generated from this project comprises data from southern Alaska (~97%). During model development, we

identified a shortcoming in GlacierCoverNet models trained on examples on the entire model dataset when classifying the northern (Brooks Range) region. These models failed to adequately capture the distribution and spatial characteristics of glaciers in northern Alaska. A simple transfer learning strategy was used to improve the capture of glaciers in the northern portions of the study area [78]. First, a GlacierCoverNet model was trained to capture glaciers in the southern portion of the study area. The weights of the initial GlacierCoverNet model were then fine-tuned on examples from the northern extent [79]. This resulted in two deep learning classifiers with different parameter values that used the same architecture and output classes. The models were applied in the northern and southern portions of the study area separately to produce maps in the final time series. These outputs were then merged for the final dataset.

GlacierCoverNet was developed using the PyTorch [80] deep learning library. The Tanimoto loss function with a complement proposed by Diakogiannis et al. [81] was used when optimizing the network. Tanimoto loss has been shown to have desirable convergence properties when semantically segmenting satellite imagery, and performs well when classes are imbalanced [81,82]. This was useful in the northern portion of the study area, where glaciers are smaller, less distinct in boundary shape, and more spatially discontinuous than in southern Alaska. The neural network weights were optimized during training using the Adam optimizer with default parameters [83] using a batch size of 64 and a learning rate of $1 \times 10^{-4}$. Both the initial GlacierCoverNet model used to map glaciers and debris-covered glaciers and the GlacierCoverNet model that was fine-tuned for identifying glaciers in the northern portions of the study area were trained using the same procedure.

### 2.5. Post-Processing Steps

The basic outputs of the GlacierCoverNet model were a full time-series (18 two-year composite images) of three-class GeoTIFFs (no glacier, debris-free glacier, and supraglacial debris). Each GeoTIFF covered the full spatial extent of Alaska and represented the medoid composite for each two-year period. These outputs did not contain any explicit information about individual glaciers or any other characteristics besides the timeframe from which they were derived and the pixel-based areas and class. The workflow for vector post-processing (Figure 2: Vector Processing) consisted of pixel-based labeling, raster-to-vector conversion, application of a minimum mapping unit (MMU), and topological correction, attribution of additional labels for unlabeled polygons and matching of RGI 6.0 attribute identification metadata.

To ensure continuity and usability for the research community, we used the RGI 6.0 glacier labels and metadata to attribute our dataset. We also used internal RGI boundaries only over ice fields (e.g., Bagley Icefield) where the boundaries between individual glaciers were not immediately clear in the classification output. In all other cases, GlacierCoverNet classifications were allowed to extend beyond or stop prior to the boundaries of RGI 6.0. To attribute glacier pixels with a label from the RGI dataset, we first labeled all GlacierCoverNet raster outputs which were spatially coincident with RGI boundaries, bypassing anywhere that RGI identified a glacier and the GlacierCoverNet outputs did not. We then applied an array-based maximum filter which used an iterative process and a $3 \times 3$ kernel to label pixels directly adjacent to already-labeled pixels. This novel region growing approach was repeated until there was less than a 1% change in new pixels labeled per iteration. This decision was based on the asymptotic behavior of the labeling process, whereby the decrease in unlabeled pixels per iteration dropped off dramatically after the first 5–10 iterations, depending on the region. This conservative approach served the dual purpose of extending labels beyond the RGI bounds and leaving some of the minor artifacts unlabeled or early/late season snow conflation present in the raw GlacierCoverNet outputs.

After labeling a majority of pixels in the raster data, we converted to vectors, removing background non-glacier pixels. Next, we applied the RGI-suggested MMU of 0.01 km$^2$ by filtering out any glacier polygons below that threshold. We then used a spatial join

process to attribute polygons which touched labeled polygons but did not inherit an RGI 6.0 label in the raster-labeling process. Finally, we used a simple attribute join to migrate the metadata from RGI 6.0 to our GlacierCoverNet output data.

### 2.6. Glacier Cover Change

To illustrate the utility of the dataset and present a novel summary of glacier change in Alaska, we characterized the overall glacier-covered area, supraglacial debris area, and overall changes with elevation and temperature shifts over the study time period. For the overall and supraglacial debris area changes, we aggregated watershed-based climate divisions, adapted from Bieniek et al. [35], to regional areas (see Section 3.1.1). To create the aggregations, we added the areas of adjacent climate divisions for each biannual composite to create the regions: Northeast Gulf, Northwest Gulf, Interior, and the Brooks Range.

To elucidate the relationship between changes in glacier-covered area and temperature over time, we used mean annual temperatures from the Oak Ridge National Lab's Daymet v4 gridded meteorological dataset, split across elevation gradients (https://daac.ornl.gov/cgi-bin/dsviewer.pl?ds_id=1904 (accessed on 28 July 2021). Daymet v4 is publicly available for 1980–2020 at a daily temporal resolution and 1000 m spatial resolution. The dataset was produced from the interpolation/extrapolation of in situ observations collected across North America [36]. To use the daily data, we first created mean annual composites in GEE that aligned with the years of GlacierCoverNet composites. We then down-sampled the GlacierCoverNet GeoTIFF outputs and IfSAR DEM to 1000 m using a cubic resampling method to match the spatial resolution of Daymet. We created 200 m elevation bins for brevity of analysis and then assigned a mean annual temperature and elevation value to each coincident pixel of the binary, down-sampled GlacierCoverNet pixels. This was conducted by multiplying binary rasters from the three datasets. We then extracted descriptive statistics from each composite for area, elevation band max, and mean annual temperature (Appendix A.3).

### 2.7. Error Analysis Reference Datasets

We adopted a multi-part approach to error analysis due to the heterogeneity of glacier-covered area in Alaska. Our error analysis consisted of four components. First, the glacier segmentation validation set (20%) was used to evaluate GlacierCoverNet's performance over spatially disjoint partitions of the modeling dataset that were withheld during training. The datasets for the southern and northern areas were evaluated independently (see Section 3.2.1). Second, we validated GlacierCoverNet's accuracy using a purpose-built point-based validation dataset (see Section 3.2.1). The point-based validation used 2000 stratified random instances of glacier-covered area interpreted from Google Earth images. This validation dataset focused on pure end member locations of debris-free glacier cover that were <100 m from distinct landcover changes (e.g., lateral moraine transitions). In creating this dataset, we endeavored to match the time stamp of the Google Earth photograph and the composite used for analysis. Third, the supraglacial debris maps produced by GlacierCoverNet were compared with the debris-covered glacier datasets created by Scherler et al. [17] and Herreid et al. [14] (see Section 3.2.2). However, the Brooks Range was not included in the mapping of supraglacial debris due to higher uncertainty in generating the class label for this region. Finally, we compared between areas of individual glaciers generated by GlacierCoverNet with RGI 6.0 (*n* = 19,490) (see Section 3.2.3). We also provide two examples from the final, classified dataset where GlacierCoverNet appears to have mapped glacier change with higher fidelity and where GlacierCoverNet overclassifies when compared with RGI 6.0 (see Section 4.2). Although a confusion matrix would generally be a helpful way of presenting some of these results, we opted not to include that here because we did not use one reference dataset for all classes for the aforementioned reasons.

## 3. Results

### 3.1. Areal Change over Time

In this section, we briefly present long-term changes and discuss some of the finer temporal changes observed in the dataset. To better understand the impact of climate on changes in glacier-covered areas, we grouped our results using watershed-based climate divisions adapted from Bieniek et al. [35], as units of analysis, and aggregated these regions by hydrologic units and geographic proximity, as outlined in Table 2. We then discuss changes in the overall glacier-covered area, area of supraglacial debris, and glacier change across temperature gradients. Finally, we evaluate GlacierCoverNet's performance and present results of the error analysis.

**Table 2.** Summary of climate division aggregations of Mann–Kandell (MK) trend test results and Theil–Sen slope ($km^2$/2 years) for aggregated regions with statistically significant trends. Additional area values are included in Appendix A.2.

| Aggregated Region | Climate Divisions ^ | MK Result * (Total Area) | MK Result * (Supraglacial Debris) | Theil–Sen Slope $km^2$/2 Years | Total Area Loss (1985–2020) ($km^2$) |
|---|---|---|---|---|---|
| Interior | Central Interior Southeast Interior | − | + | −21.2 | −1333 |
| Northeast Gulf | Northeast Gulf North Panhandle Central Panhandle South Panhandle | − | + | −125.1 | −5071 |
| Northwest Gulf | Aleutians Bristol Bay Cook Inlet Northwest Gulf | − | + | −46.4 | −2021 |
| Brooks Range | North Slope Northeast Interior | n.a. | n.a. | n.a. | n.a. |

^ List of Alaska climate divisions included in aggregated regions (Aggregated Region column) from [36]. * +, −, or n.a. denotes increasing, decreasing or not significant ($\alpha \leq 0.05$), respectively.

### 3.1.1. Overall Glacier-Covered Area

Across all the glacier-covered regions in Alaska, our data show a decline in overall glacier-covered area during the last three decades. Our results, using Mann–Kendall trend analysis (Table 2), show that all aggregated climate divisions, except for the Brooks Range, experienced a statistically significant ($\alpha \leq 0.05$) decrease in area over the study period. Across Alaska between 1985 and 2020, the overall glacier-covered area decreased by 8425 $km^2$ (−13%), although regional variability was much higher. The Northeast Gulf region has some of the largest ice-covered areas in North America, including a large proportion of the Bagley Icefield and the Bering Glacier. The Northeast Gulf region also lost the most glacier-covered area (−5071 $km^2$ or −16% change) (Figure 4) and had the largest glacier-covered area at 32,648 $km^2$ in the 1986 composite (Appendix A.2).

The smallest, statistically significant, net loss in area was in the aggregated Interior region (−1333 $km^2$ or −8%), although the aggregated Interior and Northwest Gulf regions had similar overall glacier-covered area. Despite the similar total glacier-covered area in these two regions, the Northwest Gulf lost over 65% more total area than the Interior region between 1985 and 2020.

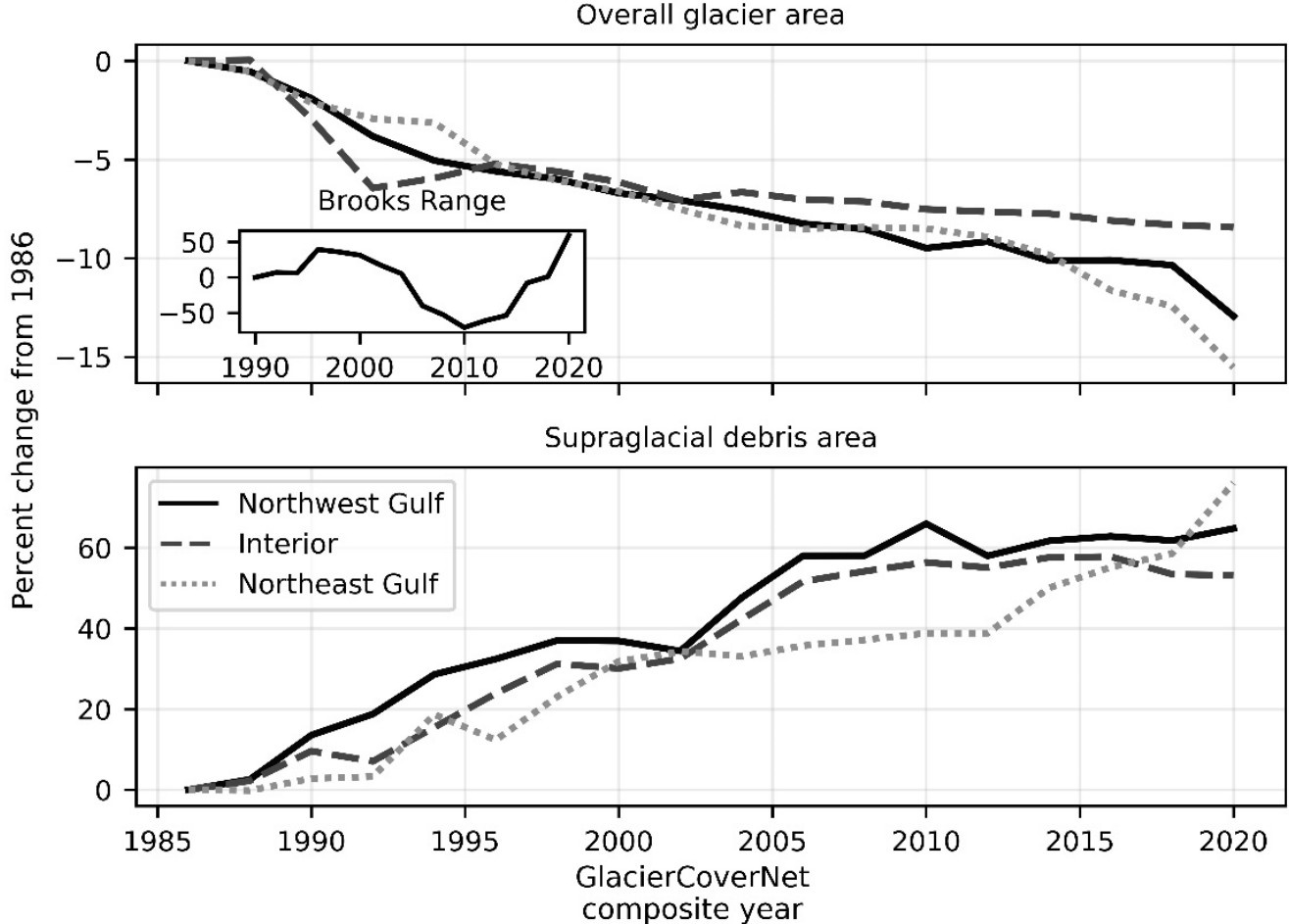

**Figure 4.** Change in overall glacier area (**top**) and area of supraglacial debris (**bottom**) for the period 1985–2020. Y-axis values are calculated as the percentage change from a 1986 composite baseline. The Brooks Range is included as an inset (**top**) and uses 1990 as a baseline due to higher uncertainty in this region, particularly early in the time series (Appendix A.2).

In addition to the long-term total changes, the structure of this dataset allows for the interrogation of some finer temporal scale trends. In all the aggregated regions, except the Brooks Range, we observed a steady or stepped decline in overall glacier-covered area during the study period. However, the greatest divergence in percentage change from the 1986 composite occurred in the last ~15 years of the study period (2005–2020). During this period, we observed a flattening, albeit a continued declining signal, in the Interior region, as compared with the first 15 years of the study period (Figure 4). In contrast, the Northwest Gulf, and especially the Northeast Gulf, experienced increasing recession as compared with the 1986 baseline in the last 5–10 years of the study (2010–2020). The Brooks Range comprises ~1% of Alaska's glacier-covered area and is spatially distant and distinct from Alaska's other glaciated regions. Both climate regions that cover the Brooks Range (Northeast Interior and North Slope) showed a marked increase in glacier-covered area in the first decade of the study period, followed by a sharp decrease with the areal minimum occurring in both regions around 2010. The period 2006–2014 produced overall areas for the Brooks Range (360 km$^2$ in 2006 to 288 km$^2$ in 2014) (Appendix A.2) that were closest to the total area recorded in RGI 6.0 (346 km$^2$ with mode year of imagery 2007). It should be noted here, for the reasons elaborated in the Discussion (Section 4), that our uncertainty is much higher in the Brooks Range than for the rest of Alaska, and this is particularly true for the time series analysis in this region. This is also reflected by the lack of statistical significance in the Mann–Kendall trend analysis (Table 2).

3.1.2. Supraglacial Debris

The aggregated glacier-covered regions of Alaska showed an increase in supraglacial debris cover over the study period, with the exception of the Brooks Range, which was not included in this part of the analysis (Table 2 and Figure 4). Across the southern part of Alaska, the area of supraglacial debris increased by 2865 km$^2$ (64%) between 1985 and 2020. The largest net increase in debris-covered area by region occurred in the Northeast Gulf by 1187 km$^2$ (76%), with the next largest increase by percentage in the aggregated Northwest Gulf 774 km$^2$, (65%) and by area in the Interior region, 904 km$^2$ or 53%. In some locations, the Southeast Panhandle, for example, we observed much higher increases (>200%).

The smallest net increase in area of supraglacial debris between 1985 and 2020 was in the Northwest Gulf (774 km$^2$), which was also the region with the smallest overall glacier-covered area of the aggregated regions in 2020 (13,604 km$^2$). Areas of supraglacial debris by biannual composite and region are included in Appendix A.2.

In the Northwest Gulf, area of supraglacial debris peaked in 2009/2010 (1983 km$^2$) before slightly decreasing or remaining stable for the remainder of the time period. Similarly, the area of supraglacial debris in the aggregated Interior region peaked in 2010–2015 (max of 2685 km$^2$ from the 2016 biannual composite) before remaining stable and then decreasing slightly at the end of the study period. Conversely, while the Northeast Gulf experienced a slower increase in supraglacial debris in the second half of the 2000–2010 decade, supraglacial debris area continued to increase until the end of the study period.

3.1.3. Changes with Elevation and Temperature

In addition to geographic location, temperature and precipitation are the primary drivers of glacier mass balance. The relationships between the accumulation of solid vs. liquid precipitation and the sensible and latent heat seasonally available for melting are key determinants of glacial mass balance. We therefore investigated how historic changes in mean annual temperature are related to glacier-covered area at different elevations (Figure 5).

Although we have established that overall glacier-covered area has declined across Alaska during the study period; recession has not been uniform with elevation gradients or years. The largest changes in overall glacier-covered area occurred in the mid-elevations of 800–2200 m, which contain the largest overall glacier-covered area. This is also consistent with our findings that the largest net glacier loss occurred in the most heavily glaciated areas of Alaska.

Steady declines in mean annual temperature values with increasing elevation are consistent with the influences of orographic cooling, sensible heat exchange, and adiabatic lapse rates. However, it is notable that in the middle and especially the last decade of the study period, mean annual temperatures increased rapidly, with the exception of 2020. This was particularly evident at the highest elevations (>3000 m), where temperatures increased by an average of ~2.5 °C between 1985 and 2020. There were some years (e.g., 2014 and 2018) with markedly larger increases (>9 °C compared with 1986). Despite these temperature changes, loss in overall glacier area at the highest elevations was minimal.

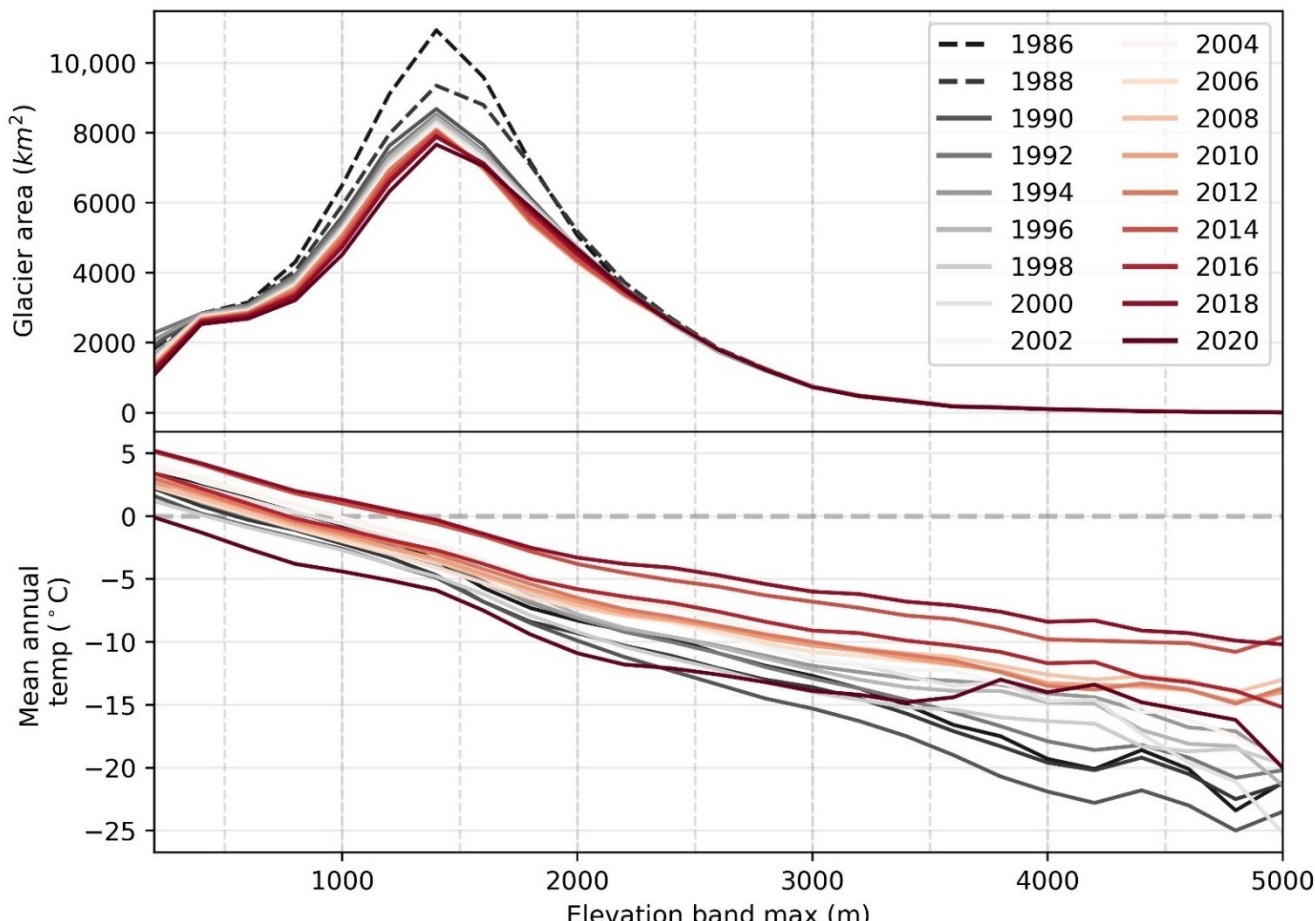

**Figure 5.** Summary of changes in glacier-covered area (**top**) and mean annual temperature (**bottom**) by biannual composite (line colors). Elevation is grouped into 200 m elevation bands. Supporting values for this figure can be found in Appendix A.3.

### 3.2. Error Analysis

### 3.2.1. Overall Glacier-Covered Area

The outputs of the GlacierCoverNet model are largely comparable to those produced through the conventional methods of band ratioing and manual editing [21] for capturing the area of debris-free glacier and supraglacial debris in the southern region of Alaska. The GlacierCoverNet outputs for 2010 (southern region) and 2006 and 2008 (northern region) were compared with the coincident years in RGI 6.0. These years were selected as the median years stated in the RGI 6.0. The median start year for RGI 6.0 in the Brooks Range is 2007; thus, we included 2006 and 2008 composites in our presentation of results. Areas for both the southern area and the Brooks Range are drawn from the 20% of 256 × 256-pixel partitions (one partition is ~59 km²) withheld from GlacierCoverNet training data generation. As compared with RGI 6.0, false positives are areas of over-classification by GlacierCoverNet, and false negatives, areas of under classification by GlacierCoverNet (Table 3).

**Table 3.** GlacierCoverNet outputs for 2010 (southern region) and 2006 and 2008 (northern region). Values are based on model outputs for areas in the validation set.

| | Southern Region | | Northern Region | | |
| --- | --- | --- | --- | --- | --- |
| | GlacierCoverNet 2010 | RGI | GlacierCoverNet 2006 | GlacierCoverNet 2008 | RGI |
| Area (km$^2$) | 10,675 | 11,030 | 94.5 | 85.4 | 110.4 |
| Area (% of RGI) | 97 | - | 86 | 77 | - |
| False positive (km$^2$) | 622.6 | - | 26.2 | 21.1 | - |
| False positive (% of RGI area) | 5.6 | - | 23.7 | 19.1 | - |
| False negative (km$^2$) | 983.7 | - | 42 | 46.1 | - |
| False negative (% of RGI area) | 8.9 | - | 38.0 | 41.8 | - |

In 2010, GlacierCoverNet identified an area of glacier cover that was 97% of the area identified in RGI 6.0 (true positive and false positive) in southern Alaska, which comprises most of the state's glacier cover (99%). The remaining, approximately 1%, of Alaska's glacier-covered area is found in the Brooks Range. In this region, GlacierCoverNet identified a total area that was 86% of RGI 6.0, which is somewhat higher than the ~10% error reported for the Brooks Range in Kienholtz et al. [21]. However, overall area statistics mask errors of omission (false negatives) and commission (false positives). In the southern region, we had 8.9% omission rates and 5.6% commission rates, largely commensurate with those reported for other studies [19,21]. The rates of omission and commission errors in the northern region were substantially higher, however, with omission rates of ~38–42% and commission rates of ~19–24%. Some of the reasons for this dissonance are included in the Discussion (Section 4).

Our analysis using the point-based validation shows similar results to the comparison with RGI 6.0 for the southern region (Table 3). We observed very high agreement (~96%) between the manually classified points and the outputs of the GlacierCoverNet model (Table 4) for the southern region. This is particularly true for the glacier-covered areas; in addition, areas of no glacier were captured with very high fidelity (i.e., minimal false positives). In the northern region, we observed higher agreement between GlacierCoverNet and the manually classified points than GlacierCoverNet and RGI 6.0 (precision = 0.93 and recall = 0.79) but a similar overall accuracy (86%).

**Table 4.** Comparison of the 2016 target year point-based validation with GlacierCoverNet for glaciers in southern Alaska and the Brooks Range.

| | Precision | Recall | F1-Score | Support |
| --- | --- | --- | --- | --- |
| Southern Region | | | | |
| 0: no glacier | 0.88 | 0.87 | 0.87 | 298 |
| 1: glacier | 0.96 | 0.96 | 0.96 | 893 |
| Accuracy | | | 0.94 | 1191 |
| Macro average | 0.92 | 0.91 | 0.92 | 1191 |
| Weighted average | 0.94 | 0.94 | 0.94 | 1191 |
| Northern Region (Brooks Range) | | | | |
| 0: no glacier | 0.81 | 0.94 | 0.87 | 211 |
| 1: glacier | 0.93 | 0.79 | 0.85 | 221 |
| Accuracy | | | 0.86 | 432 |
| Macro Average | 0.87 | 0.86 | 0.86 | 432 |
| Weighted average | 0.87 | 0.86 | 0.86 | 432 |

### 3.2.2. Supraglacial Debris Error Analysis

We analyzed the error in GlacierCoverNet supraglacial debris using two ancillary published datasets from Scherler et al. [17] and Herreid et al. [14]. We selected the year 2010 for comparison with the overall glacier-covered area results presented above, in Table 3, and 2016 to align with the stated year ranges used in Herreid et al. [14] and Scherler et al. [17] (Table 5). Similarly to the overall glacier-covered area, the supraglacial debris comparisons rely on 20% of the 256 × 256 partitions withheld from training data generation. Notably, RGI 6.0 cannot be used for supraglacial debris error analysis directly because it does not explicitly identify or calculate the area of supraglacial debris. However, it was used in the derivation of both the Scherler et al. [17] and Herreid et al. [14] datasets.

**Table 5.** Summary of area of supraglacial debris captured by GlacierCoverNet for 2010 and 2016.

| | GlacierCoverNet 2010 | GlacierCoverNet 2016 | Scherler et al. [17] | Herreid et al. [14] |
|---|---|---|---|---|
| Area ($km^2$) | 1232.8 | 1279 | 1358.8 | 1759.1 |
| Area (% of Herreid) | 70 | 73 | 77 | - |
| Area (% of Scherler) | 91 | 94 | - | 129 |
| Under classified area ($km^2$ Herreid difference) | 898.1 | 859.1 | - | - |
| Overclassified area ($km^2$ Herreid difference) | 371.5 | 378.7 | 199.4 | - |
| Under classified area ($km^2$ Scherler difference) | 701.8 | 664.2 | - | - |
| Overclassified area ($km^2$ Scherler difference) | 576 | 584.3 | - | 600 |

For both 2010 and 2016, GlacierCoverNet has high agreement with the Scherler dataset in the area of supraglacial debris (91% and 94%, respectively). For 2010 and 2016, Glacier-CoverNet also has good agreement, albeit noticeably lower, with the Herreid dataset (70% and 73%, respectively).

### 3.2.3. GlacierCoverNet vs. RGI for Individual Glaciers

Overall, the total glacier-covered areas of RGI 6.0 and GlacierCoverNet are highly comparable. However, some differences become evident when considering the size of the individual glaciers. The majority of the smallest glaciers (~<1 $km^2$) show good agreement (r = 0.78). For individual glaciers with an area above 1 $km^2$, agreement improves (r $\geq$ 0.89), and we found very high agreement between GlacierCoverNet and RGI 6.0 for the largest glaciers (r > 0.95). These relationships are shown in Figure 6, where glaciers have been extracted from the GlacierCoverNet model outputs for the year corresponding to the start year stated in the RGI 6.0 metadata. Herreid et al. [14] presented their data with an MMU of 1 $km^2$ and 2 $km^2$; however, we leave our dataset at the finer scale of the RGI-recommended 0.01 $km^2$. Thus, Figure 6, unlike Table 3, includes all glaciers for Alaska included in RGI 6.0 paired with the GlacierCoverNet result for corresponding classification years.

Some disagreement between the GlacierCoverNet outputs and the RGI 6.0 is unsurprising. In the smallest glaciers (Figure 6a). The major outliers between GlacierCoverNet and RGI 6.0 are in areas up to ~5 $km^2$ from GlacierCoverNet and are primarily located in the Brooks Range and the Aleutian chain. In addition, relatively small errors of commission have magnified effects in glaciers <5 $km^2$; the case for most of the dissonance in these small Brooks Range and Aleutian chain glaciers. These error sources are repeated, to a lesser extent, for the 1–5 $km^2$ and 5–25 $km^2$ glaciers (Figure 6b and c, respectively) and are also primarily located in the Brooks Range and Aleutian chain. Another contributing factor in the 1–25 $km^2$ glacier size is some ambiguity in the assignment of RGI 6.0 labels to GlacierCoverNet outputs, especially at the lower end of that range. For example, in cases where glaciers extend outside RGI 6.0 boundaries but have adjoining boundaries, a label must be assigned, and a boundary delineated. Errors of commission and errors in labeling

attribution can have similar disproportionate effects on glacier area because of the small overall size of these glaciers.

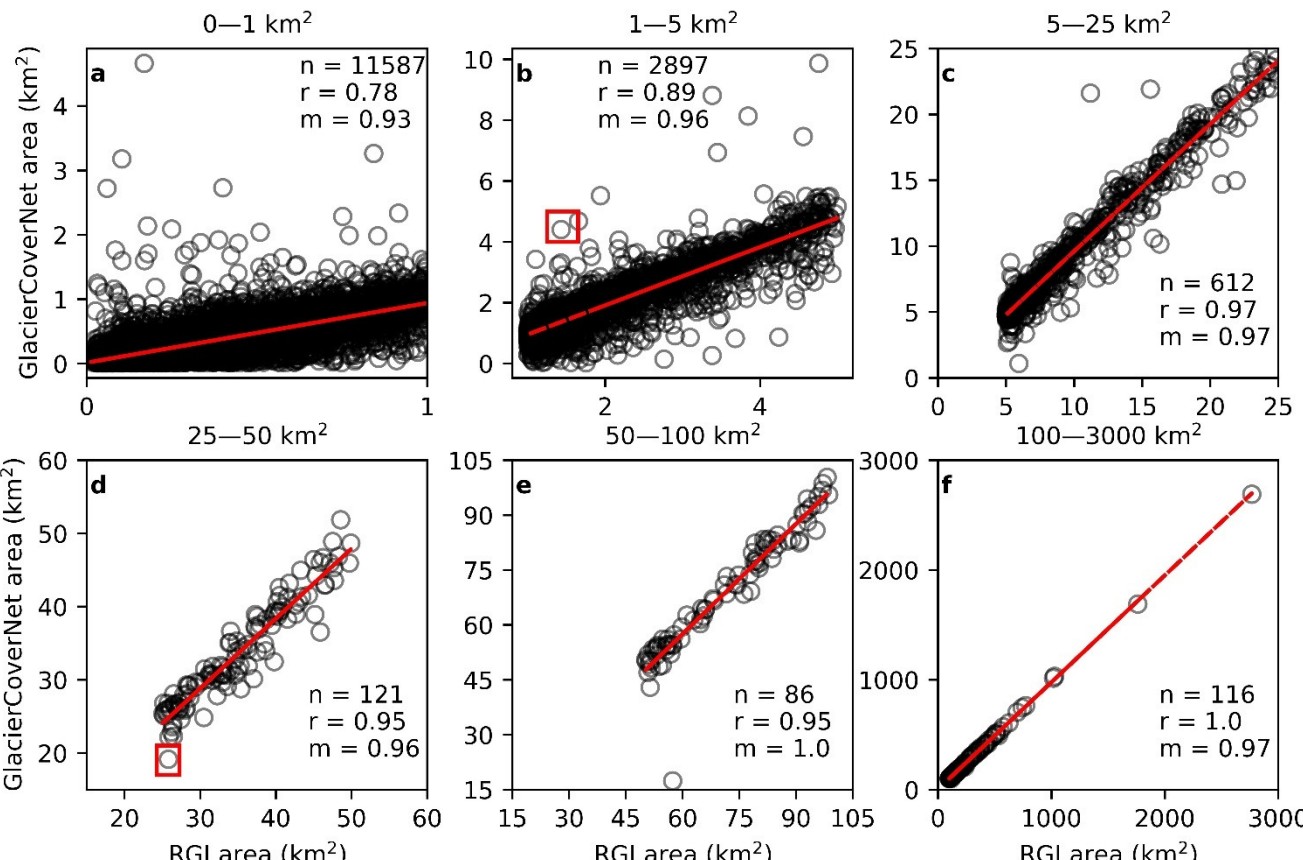

**Figure 6.** Comparison of glacier sizes from RGI 6.0 and the GlacierCoverNet model outputs with Pearson's r values and the slope (m) of the linear regression line (shown in red) for (**a**) glaciers from 0 to 1 km$^2$, (**b**) glaciers from 1 to 5 km$^2$, (**c**) glaciers from 5 to 25 km$^2$, (**d**) glaciers from 25 to 50 km$^2$, (**e**) glaciers from 50 to 100 km$^2$, and (**f**) glaciers from 100 to 3000 km$^2$.Highlighted glaciers (red box) in subplots b and d are included in Section 4.2 Figure 7 (right and left, respectively). Note that axes scales in plots a and b are intentionally different to accommodate the full range of variability in capture of smaller glaciers.

In addition to per-glacier area disagreements, of note are the small glaciers 'missing' from the GlacierCoverNet classification. In the RGI 6.0 record, there are 19,490 glaciers with unique identifiers in Alaska, but GlacierCoverNet identifies only 15,387 glaciers in 2010, a difference of ~20%. However, as noted previously, the GlacierCoverNet model outputs account for the overwhelming majority of the area identified by the RGI 6.0. The missing glaciers total 354.2 km$^2$ or ~0.6% of the overall area and have a mean area of 0.11 km$^2$ and a maximum size of 3.5 km$^2$. In contrast to 2010, in an earlier classification (1988), difference in net glacier numbers between GlacierCoverNet and RGI 6.0 drops to ~10%, a difference which represents an area of 185.6 km$^2$ (0.3% of the overall area of Alaska's glacier cover). The average individual glacier area and maximum individual glacier area of the 'missing' glaciers are analogous in 1988 as with the RGI 6.0 era.

## 4. Discussion

### 4.1. Status and Trends of Glacier-Covered Area

Alaska's glaciers are in a state of dynamic change, exacerbated by the accelerated warming of Earth's higher latitudes [84,85]. To examine these changes in glacier-covered area, and analyze their drivers and trends, we developed a novel method for creating a

multidecadal time series of biannual observations that could be adapted and expanded to use globally.

Our results indicate, in the state of Alaska over the period 1985–2020, total glacier-covered area declined by 13% and supraglacial debris cover increased by 64%. However, changes in debris-free and debris-covered glacier area are occurring at different scales and rates in different parts of Alaska, with an overall trend of decline in total area below the elevation of 2200 m (Figure 5). Although these changes have been distributed broadly over the region, the greatest changes in the southern regions have been in locations with high concentrations of tidewater and low-elevation glaciers, and the least change at the higher elevations of Alaska's Interior.

Temperature elevation gradients, from Daymet composites aggregated for glacier-covered areas, trended upward over the 1985–2020 period of study. Overall, there was an increase in biannual mean annual temperature (MAT) and a rise in the elevation of the 0 °C isotherm (Figure 5). One exception to this pattern was the 2019–2020 biannual composite (2020) that included one of the warmest summers on record (2019) and one of the coldest Decembers (2020), for Alaska, in the time frame of this analysis [86]. Although less certain, precipitation patterns over the same time period have also changed with the occurrence of more rain, less snow, and earlier spring melting. These shifts in snow accumulation and melt have been most prominent at lower elevations and during anomalously warmer years, which are occurring more frequently [87,88]. This is likely due to increasing temperatures in locations that are proximal to the Gulf of Alaska, where extreme temperatures and increasingly frequent rainfall is associated with warm air masses over the Gulf [89,90].

In response to these drivers, elevation gradients of Alaska's glacier-covered area show three distinct patterns of recession, or lack thereof, in the (i) the lowest (<400 m), (ii) middle (600–2200 m), and (iii) highest (>2200 m) elevations (Figure 5). Elevations < 400 m show a distinct incremental decline that closely corresponds with the increase in mean biannual temperature above 0 °C. The largest change in total glacier-covered area occurred in the middle elevations, specifically 600–2200 m elevations, which showed the most substantial loss in area between 1985 and 1990. These lower and mid-elevations include locations with tidewater glaciers and low-to-mid-elevation piedmont and valley glaciers and icefields in the Northeast and Northwest Gulf of Alaska, which have experienced the greatest loss in total glacier-covered area (Figure 4).

Despite the greatest increase in temperatures at the highest elevations, above 2200 m, these locations experienced little change in area (Figure 5). Glacier-covered areas above 2200 m are mostly found in the interior regions and comprise less total area in comparison to the lower elevation bands; however, changes in area do not reflect changes in volume. Furthermore, there are competing scenarios for climate at these higher elevations that are consistently below the 0 °C isotherm in MAT. Although increasing temperatures, and/or downwelling shortwave radiation would exacerbate a decrease in area and volume, increasing precipitation, as snow accumulation, would have the opposite effect. There is evidence that point to all of these scenarios; thus, this is an area of important research that may be further elucidated by future studies of mass balance and the atmospheric physics of precipitation in a warming climate.

Trends in glacier-covered area for the Brooks Range are difficult to establish due to its relatively small area and GlacierCoverNet's skill in this region (Figure 4 inset). Uncertainty in the Brooks Range is higher than in other regions of the study area due to its frequent late summer and early fall snow accumulation on these seasonally shaded cirque glaciers. Furthermore, individual glaciers in the Brooks Range glaciers are relatively small, and occur on generally steep, seasonally shaded high latitude north-facing slopes. These characteristics exacerbate the uncertainty in satellite observation and subsequent calculation of area, thus making Brooks Range glaciers inherently difficult to accurately map. When comparing GlacierCoverNet time series outputs with RGI 6.0, in the Brooks Range, the two are closely matched over the 6 years of 2007 to 2012, where RGI 6.0 has a mode of 2007 for digitized boundaries, suggesting that GlacierCoverNet is overestimating

due to conflation with seasonal snow cover (Figure 4 inset) in the early and late portions of the study period.

All regions, exclusive of Brooks Range, which were evaluated for supraglacial debris showed an increase in area and expanded across the study area by ~2800 km$^2$ (64%) throughout the study period. However, the area of supraglacial debris was not at its greatest extent in 2020, nor did its increase follow a linear trajectory. Although the less-glacier-covered locations of the Gulf of Alaska regions (e.g., Aleutians and southeast Panhandle) do not account for a large net portion of that change, their percentage increases in area of supraglacial debris were dramatic. These regions exhibited increases in supraglacial debris area of 2–4-fold over the study period. This finding has important implications for glacier mass balance studies and characterizations of surface albedo in the future.

A shift from debris-free to debris-covered surfaces can cause an increase in absorption of solar radiation and can influence melt patterns, with the direction and magnitude dependent on debris thickness [17,91–93]. The Aleutian region is dominated by active volcanoes and experienced a number of eruptions and ash or debris deposition events over the period of study (e.g., Mount Redoubt, 1989–1990 and 2009, Mount Spurr 1992, and Augustine 2005). The impact of volcanic eruptions is unrelated to climatic influence and difficult to quantify, but does impact glacier melting, through decreased albedo and increased debris cover resulting from ash deposition, lahar flows, and entrained debris. Decreased albedo, from ash, dust, and algae, has also been shown to increase glacier melt rates [94,95] and, antithetically, thick debris cover insulates the underlying ice reducing melt [96,97], the slowing of the trend in total glacier-covered area loss, as seen in Figure 5, could be a result of increased insulation and is worthy of further investigation.

### 4.2. GlacierCoverNet Performance

The time series dataset presented here has high fidelity with existing datasets of total glacier-covered area and supraglacial debris. Based on our point-based validation and the subset of the RGI 6.0, overall glacier-covered area accuracies are very high in the southern region of the state, and good in some years in the Brooks Range, albeit with much higher errors of omission and commission. Individual glacier agreement between RGI 6.0 and the GlacierCoverNet outputs was also very good for glaciers down to ~1–5 km$^2$. Glaciers < 1 km$^2$ were not captured as well, and this was one of the reasons why we found higher errors of omission and commission in the northern Brooks Range than in the southern region (e.g., Figure 7—right). We designed GlacierCoverNet to recognize the distinct geomorphological shapes of glacial lobes and associated lateral, medial, and terminal moraines; therefore, it is not surprising that the performance was decreased for the smaller, steep, shaded glaciers of the Brooks Range. Our objective was to accurately capture area change across landscape scales, which GlacierCoverNet does well, but if future studies are primarily focused on smaller, individual glaciers, the model could be adapted. Improving the capture of these small, distributed glaciers constitutes an important area of future work.

Our degree of uncertainty for supraglacial debris is higher than it is for the total glacier-covered area because of the conflation of supraglacial debris and areas of similar spectral characteristics in the proglacial environment. Additionally, for error analysis, there are limited possibilities for the validation or calibration of these estimates. The datasets that are available from Herreid et al. [14] and Scherler et al. [17] represent time periods with median imagery acquisition year of 2013 and a mode of 2015 and a range from 2013 to 2017 for Herreid et al. and Scherler et al., respectively. Due to the high degree of disparity between the Scherler and Herreid datasets, we simply labeled under-classification (error of omission) and over-classification (error of commission) to identify the disparities. It is worth noting here that Scherler et al. [17] made no modifications to the existing RGI 6.0 boundaries, whereas Herreid et al. did modify the RGI 6.0 boundaries. Herreid et al. also employed significantly more manual editing than Scherler et al.

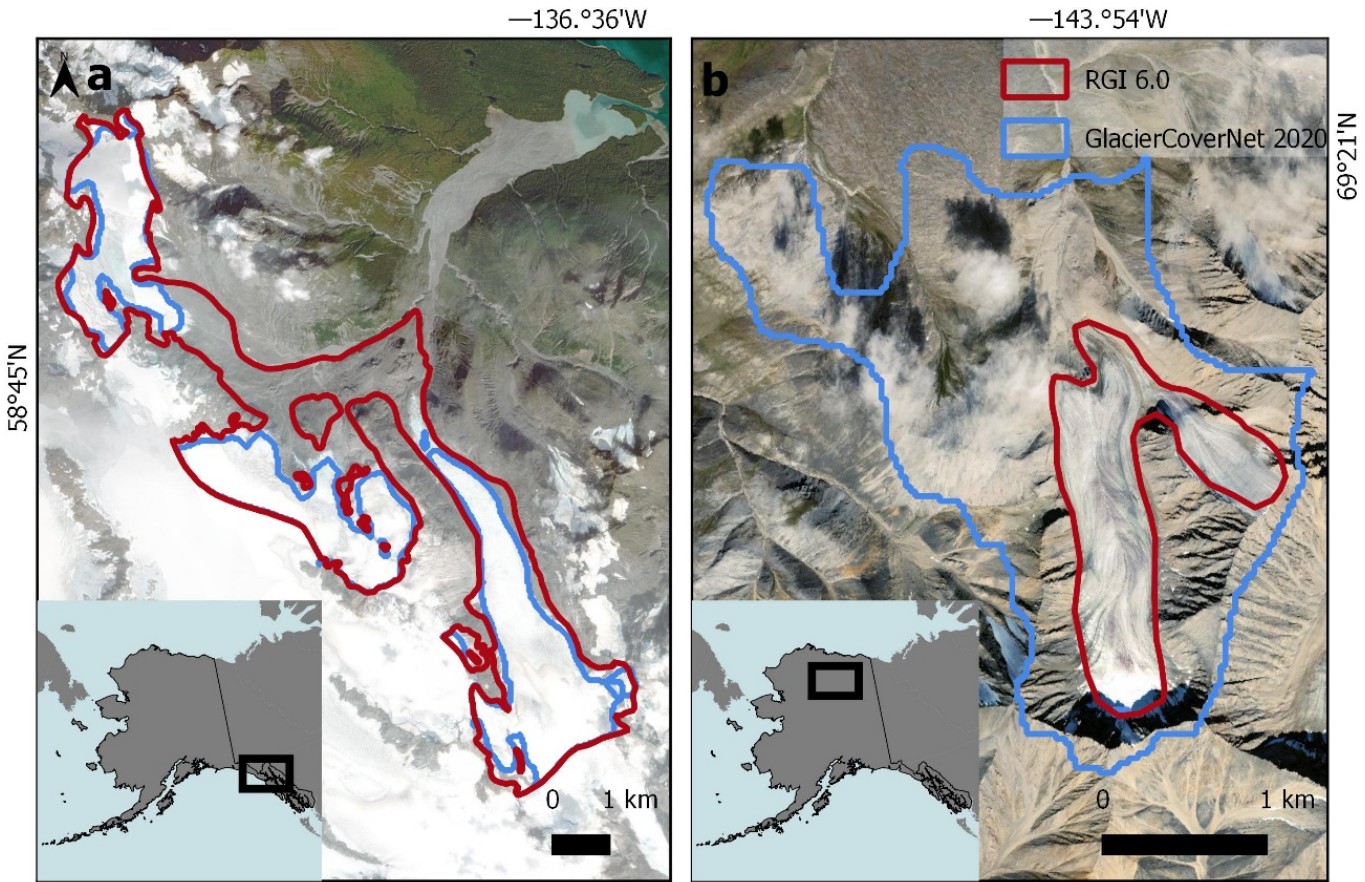

**Figure 7.** Examples of (**a**) GlacierCoverNet output (2019–2020) for the 25.8 km$^2$ Hugh Miller glacier vs. RGI 6.0 (2010) and (**b**) a small 1.4 km$^2$ unnamed glacier in the Brooks Range for GlacierCoverNet (2019–2020) vs. RGI 6.0 (2007). Note that adjacent glaciers were removed to simplify interpretation and GlacierCoverNet outputs were selected to align with base imagery. Underlying imagery is from the Environmental Systems Research Institute (ESRI) World Imagery layer, 15 m satellite imagery from TerraColor NextGen (https://www.arcgis.com/home/item.html?id=10df2279f9684e4a9f6a7f0 8febac2a9 (accessed on 1 August 2022)).

Much of the uncertainty we found in identifying small glaciers is likely attributable to conflation by GlacierCoverNet between glacier ice and out of season snow cover (Figure 7—right). This led to a tendency of GlacierCoverNet to overclassify glacier-covered area in the Brooks Range where the snow-free period is generally short and can be present at any time of the year, and the Aleutian chain where there is persistent cloud cover and, at times, sparse coverage from satellite overpasses. Conversely, we found that our approach was also able to produce glacier outlines that appear to better represent in situ conditions than existing RGI 6.0 data (e.g., Figure 7—left). It should be noted that outlines from RGI (up to 6.0), in most cases, are acquired from a single date; thus, capturing a single snapshot in time. In contrast, GlacierCoverNet provides a biannual medoid time series that allows for the quantification of regional change or trends using a common and repeatable methodology.

### 4.3. Uncertainties and Future Work

Some sources of uncertainty remain in the workflow and outputs of GlacierCover-Net which, if addressed, will improve its broader application. In the workflow methodology, the classification algorithms used are limited by the input training data and, in some cases, by the model's architecture. Issues with the training data and target layer have two main considerations: (1) data for predictor variables and (2) class labels. As noted, we selected the partition sizes to build the training dataset largely based on the

geomorphological features that define Alaska's southern glaciated landscape with an emphasis on improving the capture of debris-covered glacier lobes and glacier termini. The drawback of that choice in terms of model design was that GlacierCoverNet was less accurate with small glaciers < 1 km$^2$ and, in some cases, up to 5 km$^2$ (Figure 6). Although the Brooks Range comprised only about 1% of the total area analyzed in our study, we paid particular attention to this region for its unique qualities that test the limits of GlacierCoverNet's transferability. We trained the model that was deployed for the Brooks Range glaciers on Alaska's southern region, which limited our ability to accurately map the predominantly small, discontinuous, shaded, high-latitude glaciers of the Brooks Range. This could be improved by shifting the emphasis or expanding the scope of the training dataset and associated model weights and architecture to improve capture of these types of glaciers.

The largest uncertainties in the optical data that were used to train GlacierCoverNet resulted from data loss in the available Landsat images and the spectral similarity of out-of-season snow and glacier-covered areas and supraglacial debris and the surrounding terrain. Landsat coverage over Alaska in the 1990s was limited due to the launch failure of Landsat 6, and especially data loss due to limited downlinking over this region during this period. Although additional optical data are largely unavailable for this epoch, it may be possible to augment processing with the data of different spatial granularity or source instruments in the future. Cloud contamination also created data gaps that were filled programmatically using the LT algorithm, but longer gaps without valid imagery also resulted in higher uncertainty. Although our cloud mask was effective, recent improvements in cloud masking [98] could be integrated into future versions of the workflow. As discussed above, the conflations of snow with glacier ice, firn, and névé confound optical imagery glacier classification. The date window for selecting the best optical data could be further improved by adopting region- and/or elevation-specific windows and/or by the inclusion of ancillary snow cover data from other satellite sensors.

The high spatial resolution (5 m) IfSAR is the only high-quality continuous elevation product that includes all of our study area, but it is limited to the state of Alaska. Furthermore, the topographic indices produced from these data are derived from acquisitions collected in the 2014–2017 period and are thus temporally static. This means that large topographic features, calving faces at the toe of a glacier for example, are only captured for that snapshot in time. In the future, a time series tracking the progression of glacial feature predictor variables could potentially be generated from alternate airborne or spaceborne sources such as interferometry or LiDAR and may become available in future versions of the Global Land Ice Measurements from Space (GLIMS) (https://www.glims.org/RGI/, accessed on 1 August 2022).

There are additional potential sources of uncertainty in the class label which we attempt to address in our workflow. Although the RGI 6.0 data are crucial for this project, there are inherent uncertainties even in manually edited glacier outlines [14,21]. These uncertainties stem from differences in observer interpretations and human error, in addition to the issues with optical imagery, as outlined above (e.g., shadows and a lack of imagery of adequate spatial resolution for resolving highly heterogeneous areas). There is also inherent ambiguity in the RGI 6.0 because of the multiple dates of imagery selected for boundary delineation. Despite these uncertainties, the RGI 6.0 was integral to our model training dataset; its use dramatically decreased the time that would have been required to manually build a large training dataset and enabled us to create an output dataset in a relatively short period of time. We also used the RGI 6.0 labels to inform the identification of our output classifications which also removes much of the GlacierCoverNet model output which is out-of-season snow misclassified as glacier ice. Lastly, future versions of RGI will likely provide additional opportunities for data assimilation and validation.

The STEM model had 95% overall model accuracy based on our point-based validation and performed as well or better than other models attempting to map debris-free glacier cover (e.g., band ratioing); however, as with any classification method, the classi-

fication of debris-free ice and supraglacial debris will contain some inherent error. The uncertainty in this class label also introduces uncertainty into the final GlacierCoverNet modeling process and outputs. Some of the steps we took to address these issues are discussed in the Methods section above, but further improvement could be made to the class label by introducing more regionally specific classification approaches or by more editing of initial debris-free ice outputs [99]. A band ratioing or other similar less computationally intensive approach would also likely be required if applying the GlacierCoverNet model to a global scope.

## 5. Conclusions

The methods and dataset introduced here represent an important step in the automated capture of glacier change. Our approach builds on existing work to produce data at a higher temporal frequency with less manual editing than other methods currently in use and will hopefully inform future studies in this area of research. Although our geographic focus was on the state of Alaska, the methods and modeling approach was built to be scalable and flexible. Based on our analysis of results and errors, we believe that this model will perform well in landscapes with a similar glacial makeup (e.g., coastal British Columbia). However, due to the high uncertainty in the Brooks Range, our approach to training and model architecture will require additional development for areas dominated by shaded terrain and small mountain glaciers. This research has the potential to expand our ability to capture changes and evaluate trends in this important component of the global cryosphere.

**Author Contributions:** Conceptualization, B.M.R.-P., P.B.K., R.E.K. and J.B.K.; methodology, B.M.R.-P., P.B.K. and J.B.K.; software, B.M.R.-P. and J.B.K.; validation, B.M.R.-P. and J.B.K.; formal analysis, B.M.R.-P. and P.B.K.; investigation, B.M.R.-P. and P.B.K.; data curation, B.M.R.-P.; writing—original draft preparation, B.M.R.-P., P.B.K. and J.B.K.; writing—review and editing, B.M.R.-P., P.B.K., J.B.K. and R.E.K.; visualization, B.M.R.-P.; supervision, P.B.K. and R.E.K.; project administration, P.B.K. and R.E.K.; funding acquisition, P.B.K. All authors have read and agreed to the published version of the manuscript.

**Funding:** This study was supported by the National Park Service, Inventory & Monitoring Program, Southwest Alaska Network and Focused Condition Funds, through a Cooperative Ecosystems Studies Unit agreement awarded to R. Kennedy (P20AC00176). Publication of these data was made possible by NCEI support to NOAA@NSIDC through the NOAA Cooperative Agreement with CIRES, NA17OAR4320101.

**Data Availability Statement:** The dataset associated with this study is available at the National Snow and Ice Data Center (NSIDC) and can be cited as: Roberts-Pierel, B.M., Kirchner, P.B., Kilbride, J.B., Kennedy, R.E. 2022. Glacier-Covered Area for the State of Alaska, 1985–2020, Version 1. [Indicate subset used]. Boulder, Colorado USA. NSIDC: National Snow and Ice Data Center. (https://doi.org/10.7265/8esq-w553, accessed on 1 August 2022).

**Acknowledgments:** We would like to thank and acknowledge the three anonymous reviewers who provided helpful insight and suggestions to the review and editing of this manuscript. Their input certainly improved the clarity and value of this publication.

**Conflicts of Interest:** The authors declare no conflict of interest.

**Software Statement:** All code associated with this study is currently available from the corresponding author upon request.

# Appendix A.

*Appendix A.1. GlacierCoverNet Architecture*

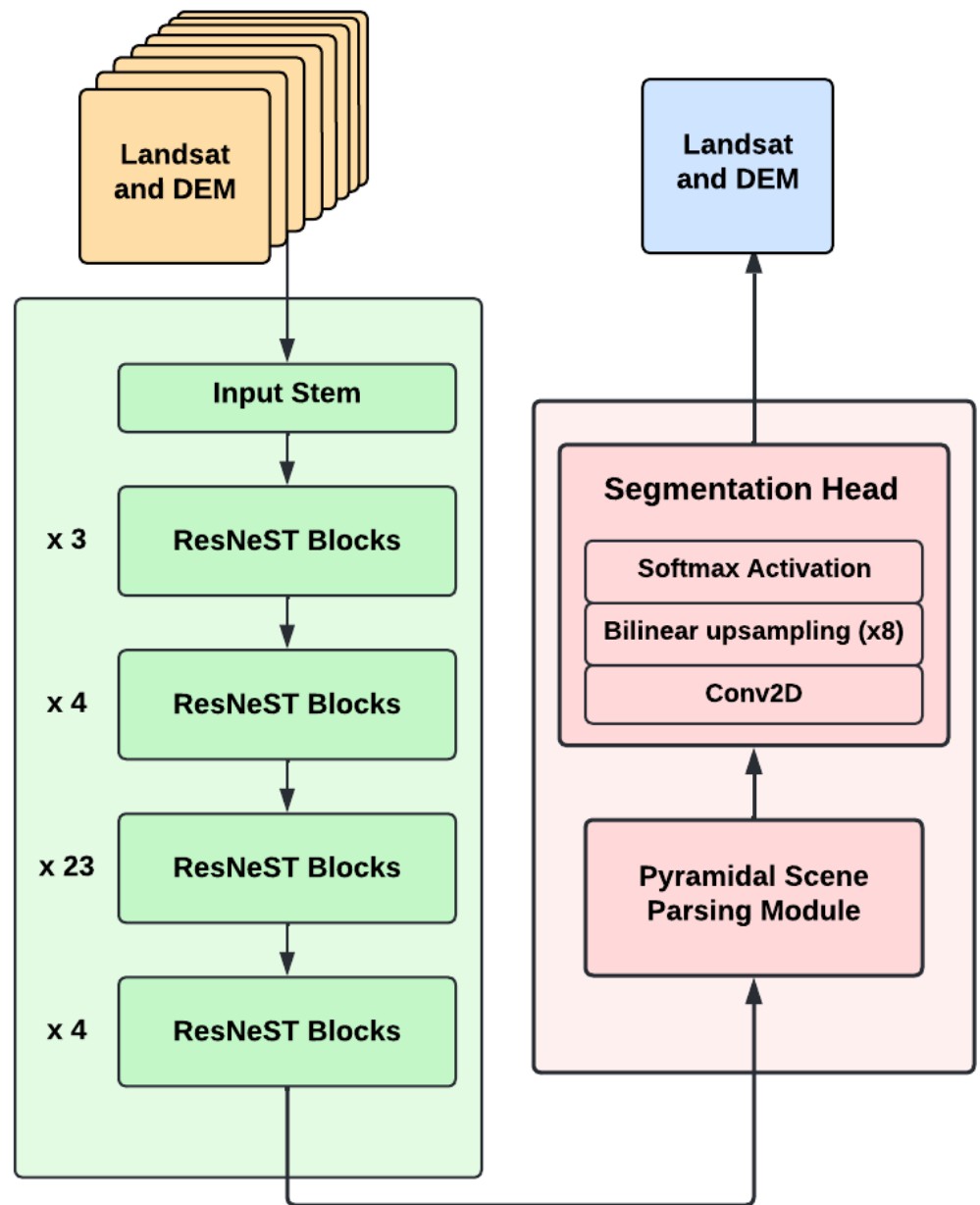

**Figure A1.** Conceptual diagram of GlacierCoverNet model architecture and processing scheme.

*Appendix A.2. Glacier-Covered Area Values*

**Table A1.** Overall glacier-covered area for biannual composites for aggregated climate divisions. Note that areas for the Brooks Range are excluded from total areas and percentage changes due to higher uncertainty in this region. Areas are km$^2$.

| Composite Year | Northwest Gulf | Northeast Gulf | Interior | Brooks Range | Total |
|---|---|---|---|---|---|
| 1986 | 15,625.1 | 32,648.3 | 15,803.5 | - | 64,076.9 |
| 1988 | 15,541.4 | 32,469.7 | 15,811.6 | - | |
| 1990 | 15,328.9 | 31,957.7 | 15,337.2 | 565.7 | |
| 1992 | 15,028.4 | 31,692.7 | 14,784.8 | 604.3 | |
| 1994 | 14,836.1 | 31,624.9 | 14,864.2 | 600.2 | |
| 1996 | 14,751.5 | 30,938.5 | 14,980.2 | 788.0 | |
| 1998 | 14,689.0 | 30,670.4 | 14,917.1 | 768.4 | |
| 2000 | 14,578.4 | 30,491.0 | 14,833.7 | 742.8 | |
| 2002 | 14,522.0 | 30,194.1 | 14,690.7 | 663.1 | |
| 2004 | 14,444.4 | 29,921.9 | 14,752.8 | 594.8 | |
| 2006 | 14,336.5 | 29,870.9 | 14,694.7 | 339.1 | |
| 2008 | 14,293.6 | 29,896.6 | 14,677.3 | 267.1 | |
| 2010 | 14,143.4 | 29,874.1 | 14,615.7 | 166.2 | |
| 2012 | 14,195.0 | 29,735.5 | 14,596.1 | 220.9 | |
| 2014 | 14,042.8 | 29,442.0 | 14,580.4 | 262.0 | |
| 2016 | 14,048.0 | 28,856.5 | 14,525.3 | 520.9 | |
| 2018 | 14,008.4 | 28,591.5 | 14,491.4 | 571.4 | |
| 2020 | 13,604.1 | 27,577.2 | 14,470.9 | 901.5 | 55,652.2 |
| Net Area Change | −2021.0 | −5071.1 | −1332.6 | 335.9 | −8424.7 |
| % Change | −12.9 | −15.5 | −8.4 | 59.4 | −13.1 |

**Table A2.** Supraglacial debris area by biannual composite and aggregated climate divisions. Areas are km$^2$.

| Composite Year | Northwest Gulf | Northeast Gulf | Interior | Total |
|---|---|---|---|---|
| 1986 | 1195.1 | 1560.5 | 1701.8 | 4457.3 |
| 1988 | 1225.7 | 1557.7 | 1739.7 | |
| 1990 | 1356.9 | 1604.1 | 1865.0 | |
| 1992 | 1419.2 | 1611.2 | 1821.9 | |
| 1994 | 1536.8 | 1851.9 | 1965.3 | |
| 1996 | 1582.3 | 1753.4 | 2108.2 | |
| 1998 | 1637.4 | 1920.7 | 2233.2 | |
| 2000 | 1636.0 | 2057.1 | 2213.7 | |
| 2002 | 1606.1 | 2095.0 | 2254.6 | |
| 2004 | 1764.1 | 2076.7 | 2420.2 | |
| 2006 | 1888.4 | 2117.7 | 2580.7 | |
| 2008 | 1888.5 | 2140.2 | 2623.4 | |
| 2010 | 1983.1 | 2166.0 | 2660.9 | |
| 2012 | 1887.8 | 2166.0 | 2638.9 | |
| 2014 | 1932.4 | 2341.2 | 2682.0 | |
| 2016 | 1945.7 | 2420.9 | 2685.0 | |
| 2018 | 1933.2 | 2473.5 | 2611.0 | |
| 2020 | 1969.0 | 2747.3 | 2606.1 | 7322.4 |
| Net area change | 773.9 | 1186.9 | 904.2 | 2865.0 |
| % Change | 64.8 | 76.1 | 53.1 | 64.3 |

*Appendix A.3. Change in Glacier-Covered Area with Mean Annual Temperature and Elevation*

**Table A3.** Summary of overall glacier-covered area by biannual composite, 200 m elevation band, and mean annual temperature; elevation band label is the upper extent.

| Year | Elevation Band (m) | Area (km²) | Mean Annual Temp (°C) | Standard Deviation |
|------|------|------|------|------|
| 1986 | 200 | 1617 | 3.6 | 1.0 |
| 1986 | 400 | 2836 | 2.4 | 1.1 |
| 1986 | 600 | 3155 | 1.4 | 1.4 |
| 1986 | 800 | 4308 | 0.3 | 1.8 |
| 1986 | 1000 | 6510 | −0.7 | 2.2 |
| 1986 | 1200 | 9108 | −1.9 | 2.6 |
| 1986 | 1400 | 10,936 | −3.5 | 3.3 |
| 1986 | 1600 | 9595 | −5.7 | 3.9 |
| 1986 | 1800 | 7159 | −7.3 | 3.9 |
| 1986 | 2000 | 5066 | −8.3 | 3.7 |
| 1986 | 2200 | 3551 | −9.1 | 3.1 |
| 1986 | 2400 | 2574 | −9.7 | 2.5 |
| 1986 | 2600 | 1764 | −10.9 | 2.2 |
| 1986 | 2800 | 1215 | −11.9 | 2.3 |
| 1986 | 3000 | 735 | −12.7 | 2.6 |
| 1986 | 3200 | 473 | −13.7 | 2.8 |
| 1986 | 3400 | 343 | −15.0 | 3.0 |
| 1986 | 3600 | 183 | −16.6 | 2.8 |
| 1986 | 3800 | 151 | −17.5 | 3.3 |
| 1986 | 4000 | 104 | −19.3 | 3.4 |
| 1986 | 4200 | 79 | −20.1 | 3.8 |
| 1986 | 4400 | 47 | −18.6 | 2.8 |
| 1986 | 4600 | 28 | −20.1 | 3.6 |
| 1986 | 4800 | 19 | −23.4 | 2.3 |
| 1986 | 5000 | 7 | −21.2 | 3.4 |
| 1988 | 200 | 1822 | 2.2 | 1.0 |
| 1988 | 400 | 2819 | 0.8 | 1.2 |
| 1988 | 600 | 3097 | −0.3 | 1.5 |
| 1988 | 800 | 4055 | −1.1 | 1.8 |
| 1988 | 1000 | 5932 | −2.2 | 2.0 |
| 1988 | 1200 | 7968 | −3.3 | 2.3 |
| 1988 | 1400 | 9351 | −4.7 | 3.0 |
| 1988 | 1600 | 8800 | −6.8 | 3.5 |
| 1988 | 1800 | 7083 | −8.4 | 3.2 |
| 1988 | 2000 | 5184 | −9.3 | 2.8 |
| 1988 | 2200 | 3731 | −10.3 | 2.2 |
| 1988 | 2400 | 2697 | −11.1 | 1.7 |
| 1988 | 2600 | 1837 | −12.1 | 1.6 |
| 1988 | 2800 | 1262 | −13.0 | 1.6 |
| 1988 | 3000 | 754 | −13.6 | 1.8 |
| 1988 | 3200 | 492 | −14.5 | 1.9 |
| 1988 | 3400 | 351 | −15.7 | 2.1 |
| 1988 | 3600 | 185 | −17.1 | 2.0 |
| 1988 | 3800 | 154 | −18.3 | 2.1 |
| 1988 | 4000 | 104 | −19.6 | 2.3 |
| 1988 | 4200 | 79 | −20.2 | 2.4 |
| 1988 | 4400 | 47 | −19.2 | 1.9 |
| 1988 | 4600 | 28 | −20.5 | 1.8 |
| 1988 | 4800 | 19 | −22.5 | 1.8 |
| 1988 | 5000 | 7 | −21.3 | 0.8 |

**Table A3.** *Cont.*

| Year | Elevation Band (m) | Area (km²) | Mean Annual Temp (°C) | Standard Deviation |
|------|------|------|------|------|
| 1990 | 200  | 1931 | 1.6   | 1.0 |
| 1990 | 400  | 2828 | 0.2   | 1.2 |
| 1990 | 600  | 3080 | −0.8  | 1.5 |
| 1990 | 800  | 3930 | −1.7  | 1.9 |
| 1990 | 1000 | 5609 | −2.6  | 2.2 |
| 1990 | 1200 | 7631 | −3.8  | 2.4 |
| 1990 | 1400 | 8684 | −4.9  | 2.5 |
| 1990 | 1600 | 7657 | −6.8  | 2.7 |
| 1990 | 1800 | 6132 | −8.5  | 2.5 |
| 1990 | 2000 | 4712 | −9.9  | 2.3 |
| 1990 | 2200 | 3568 | −11.2 | 2.0 |
| 1990 | 2400 | 2641 | −12.3 | 1.9 |
| 1990 | 2600 | 1808 | −13.4 | 2.1 |
| 1990 | 2800 | 1249 | −14.5 | 2.1 |
| 1990 | 3000 | 760  | −15.3 | 2.2 |
| 1990 | 3200 | 490  | −16.3 | 2.1 |
| 1990 | 3400 | 354  | −17.5 | 2.4 |
| 1990 | 3600 | 186  | −19.0 | 2.4 |
| 1990 | 3800 | 153  | −20.7 | 2.2 |
| 1990 | 4000 | 104  | −21.9 | 2.4 |
| 1990 | 4200 | 77   | −22.8 | 2.5 |
| 1990 | 4400 | 47   | −21.8 | 2.2 |
| 1990 | 4600 | 28   | −23.0 | 2.2 |
| 1990 | 4800 | 18   | −25.0 | 2.6 |
| 1990 | 5000 | 7    | −23.5 | 0.9 |
| 1992 | 200  | 2280 | 3.0   | 0.9 |
| 1992 | 400  | 2834 | 1.6   | 1.0 |
| 1992 | 600  | 3074 | 0.5   | 1.3 |
| 1992 | 800  | 3919 | −0.4  | 1.7 |
| 1992 | 1000 | 5484 | −1.3  | 1.9 |
| 1992 | 1200 | 7411 | −2.4  | 2.1 |
| 1992 | 1400 | 8523 | −3.5  | 2.4 |
| 1992 | 1600 | 7476 | −5.2  | 2.6 |
| 1992 | 1800 | 5995 | −6.9  | 2.6 |
| 1992 | 2000 | 4623 | −8.1  | 2.3 |
| 1992 | 2200 | 3436 | −9.2  | 2.0 |
| 1992 | 2400 | 2525 | −10.0 | 1.6 |
| 1992 | 2600 | 1741 | −10.9 | 1.5 |
| 1992 | 2800 | 1198 | −12.0 | 1.5 |
| 1992 | 3000 | 729  | −12.9 | 1.5 |
| 1992 | 3200 | 475  | −13.7 | 1.4 |
| 1992 | 3400 | 347  | −14.6 | 1.6 |
| 1992 | 3600 | 180  | −15.6 | 1.5 |
| 1992 | 3800 | 150  | −16.7 | 1.6 |
| 1992 | 4000 | 104  | −17.9 | 1.8 |
| 1992 | 4200 | 76   | −18.6 | 1.7 |
| 1992 | 4400 | 45   | −18.2 | 1.9 |
| 1992 | 4600 | 28   | −19.2 | 1.5 |
| 1992 | 4800 | 18   | −20.8 | 2.1 |
| 1992 | 5000 | 7    | −20.2 | 1.4 |
| 1994 | 200  | 2083 | 2.6   | 1.0 |
| 1994 | 400  | 2817 | 1.1   | 1.2 |
| 1994 | 600  | 3028 | 0.0   | 1.4 |
| 1994 | 800  | 3822 | −1.0  | 1.6 |

**Table A3.** *Cont.*

| Year | Elevation Band (m) | Area (km²) | Mean Annual Temp (°C) | Standard Deviation |
|---|---|---|---|---|
| 1994 | 1000 | 5368 | −2.0 | 1.6 |
| 1994 | 1200 | 7311 | −3.0 | 1.9 |
| 1994 | 1400 | 8490 | −4.0 | 2.1 |
| 1994 | 1600 | 7418 | −5.3 | 2.4 |
| 1994 | 1800 | 5981 | −6.8 | 2.4 |
| 1994 | 2000 | 4643 | −8.1 | 2.1 |
| 1994 | 2200 | 3439 | −9.0 | 1.8 |
| 1994 | 2400 | 2539 | −9.6 | 1.6 |
| 1994 | 2600 | 1746 | −10.3 | 1.7 |
| 1994 | 2800 | 1233 | −11.1 | 2.0 |
| 1994 | 3000 | 745 | −11.9 | 2.0 |
| 1994 | 3200 | 486 | −12.4 | 2.1 |
| 1994 | 3400 | 348 | −12.9 | 2.1 |
| 1994 | 3600 | 181 | −13.1 | 2.2 |
| 1994 | 3800 | 152 | −13.4 | 2.2 |
| 1994 | 4000 | 104 | −14.1 | 2.3 |
| 1994 | 4200 | 78 | −14.4 | 2.3 |
| 1994 | 4400 | 47 | −15.5 | 1.8 |
| 1994 | 4600 | 28 | −16.8 | 2.1 |
| 1994 | 4800 | 19 | −17.1 | 2.2 |
| 1994 | 5000 | 7 | −19.7 | 1.1 |
| 1996 | 200 | 1602 | 3.1 | 1.1 |
| 1996 | 400 | 2790 | 1.9 | 1.3 |
| 1996 | 600 | 2971 | 0.7 | 1.5 |
| 1996 | 800 | 3740 | −0.2 | 1.9 |
| 1996 | 1000 | 5316 | −1.1 | 2.1 |
| 1996 | 1200 | 7272 | −2.1 | 2.3 |
| 1996 | 1400 | 8431 | −3.0 | 2.5 |
| 1996 | 1600 | 7425 | −4.5 | 2.9 |
| 1996 | 1800 | 6042 | −6.2 | 3.1 |
| 1996 | 2000 | 4741 | −7.8 | 3.0 |
| 1996 | 2200 | 3510 | −8.9 | 2.6 |
| 1996 | 2400 | 2569 | −9.6 | 1.9 |
| 1996 | 2600 | 1786 | −10.4 | 1.7 |
| 1996 | 2800 | 1240 | −11.3 | 1.8 |
| 1996 | 3000 | 753 | −12.2 | 1.9 |
| 1996 | 3200 | 487 | −13.0 | 2.0 |
| 1996 | 3400 | 343 | −13.6 | 2.2 |
| 1996 | 3600 | 182 | −13.9 | 2.5 |
| 1996 | 3800 | 150 | −13.9 | 2.3 |
| 1996 | 4000 | 104 | −14.8 | 2.8 |
| 1996 | 4200 | 78 | −14.9 | 2.8 |
| 1996 | 4400 | 48 | −17.0 | 2.4 |
| 1996 | 4600 | 28 | −18.1 | 2.7 |
| 1996 | 4800 | 19 | −18.3 | 3.5 |
| 1996 | 5000 | 7 | −21.4 | 2.3 |
| 1998 | 200 | 1599 | 1.2 | 1.2 |
| 1998 | 400 | 2766 | 0.1 | 1.3 |
| 1998 | 600 | 2941 | −0.9 | 1.5 |
| 1998 | 800 | 3696 | −1.8 | 1.8 |
| 1998 | 1000 | 5284 | −2.7 | 1.9 |
| 1998 | 1200 | 7155 | −3.8 | 2.2 |
| 1998 | 1400 | 8324 | −4.8 | 2.3 |
| 1998 | 1600 | 7307 | −6.2 | 2.3 |

**Table A3.** *Cont.*

| Year | Elevation Band (m) | Area (km²) | Mean Annual Temp (°C) | Standard Deviation |
|------|------|------|------|------|
| 1998 | 1800 | 5965 | −7.9 | 2.1 |
| 1998 | 2000 | 4703 | −9.2 | 1.9 |
| 1998 | 2200 | 3489 | −10.4 | 1.6 |
| 1998 | 2400 | 2564 | −11.3 | 1.7 |
| 1998 | 2600 | 1777 | −12.2 | 1.8 |
| 1998 | 2800 | 1230 | −13.2 | 2.0 |
| 1998 | 3000 | 746 | −14.0 | 2.0 |
| 1998 | 3200 | 486 | −14.6 | 2.1 |
| 1998 | 3400 | 347 | −15.2 | 2.3 |
| 1998 | 3600 | 183 | −15.4 | 2.3 |
| 1998 | 3800 | 152 | −16.0 | 2.7 |
| 1998 | 4000 | 102 | −16.3 | 2.8 |
| 1998 | 4200 | 78 | −16.5 | 3.0 |
| 1998 | 4400 | 48 | −18.3 | 2.2 |
| 1998 | 4600 | 28 | −18.7 | 2.2 |
| 1998 | 4800 | 19 | −18.5 | 1.6 |
| 1998 | 5000 | 7 | −19.7 | 1.9 |
| 2000 | 200 | 1514 | 3.3 | 0.9 |
| 2000 | 400 | 2738 | 2.3 | 0.9 |
| 2000 | 600 | 2901 | 1.3 | 1.2 |
| 2000 | 800 | 3657 | 0.3 | 1.4 |
| 2000 | 1000 | 5218 | −0.6 | 1.4 |
| 2000 | 1200 | 7105 | −1.7 | 1.6 |
| 2000 | 1400 | 8283 | −2.9 | 1.9 |
| 2000 | 1600 | 7282 | −4.5 | 2.4 |
| 2000 | 1800 | 5933 | −6.1 | 2.9 |
| 2000 | 2000 | 4666 | −7.1 | 2.9 |
| 2000 | 2200 | 3469 | −7.6 | 2.6 |
| 2000 | 2400 | 2569 | −8.2 | 2.2 |
| 2000 | 2600 | 1794 | −9.2 | 2.4 |
| 2000 | 2800 | 1246 | −10.3 | 2.7 |
| 2000 | 3000 | 757 | −11.4 | 3.1 |
| 2000 | 3200 | 495 | −11.8 | 3.0 |
| 2000 | 3400 | 351 | −12.7 | 3.2 |
| 2000 | 3600 | 187 | −13.5 | 3.6 |
| 2000 | 3800 | 151 | −13.2 | 3.1 |
| 2000 | 4000 | 103 | −14.7 | 4.2 |
| 2000 | 4200 | 78 | −14.6 | 3.9 |
| 2000 | 4400 | 48 | −17.3 | 4.1 |
| 2000 | 4600 | 28 | −19.6 | 5.0 |
| 2000 | 4800 | 19 | −21.1 | 6.0 |
| 2000 | 5000 | 7 | −25.2 | 3.7 |
| 2002 | 200 | 1464 | 3.7 | 1.0 |
| 2002 | 400 | 2733 | 2.8 | 0.9 |
| 2002 | 600 | 2914 | 1.8 | 1.1 |
| 2002 | 800 | 3638 | 0.9 | 1.3 |
| 2002 | 1000 | 5175 | −0.1 | 1.4 |
| 2002 | 1200 | 7016 | −1.2 | 1.5 |
| 2002 | 1400 | 8207 | −2.2 | 1.7 |
| 2002 | 1600 | 7181 | −3.6 | 2.0 |
| 2002 | 1800 | 5857 | −5.2 | 2.4 |
| 2002 | 2000 | 4576 | −6.5 | 2.4 |
| 2002 | 2200 | 3422 | −7.6 | 2.3 |
| 2002 | 2400 | 2547 | −8.5 | 2.2 |

**Table A3.** *Cont.*

| Year | Elevation Band (m) | Area (km²) | Mean Annual Temp (°C) | Standard Deviation |
|---|---|---|---|---|
| 2002 | 2600 | 1784 | −9.4 | 2.4 |
| 2002 | 2800 | 1240 | −10.4 | 2.5 |
| 2002 | 3000 | 752 | −11.3 | 2.7 |
| 2002 | 3200 | 496 | −11.8 | 2.8 |
| 2002 | 3400 | 353 | −12.2 | 2.8 |
| 2002 | 3600 | 190 | −12.3 | 2.8 |
| 2002 | 3800 | 149 | −13.3 | 3.1 |
| 2002 | 4000 | 103 | −13.4 | 3.3 |
| 2002 | 4200 | 78 | −13.5 | 3.5 |
| 2002 | 4400 | 47 | −15.6 | 3.7 |
| 2002 | 4600 | 27 | −15.4 | 3.5 |
| 2002 | 4800 | 19 | −14.6 | 2.4 |
| 2002 | 5000 | 8 | −14.5 | 2.5 |
| 2004 | 200 | 1450 | 4.1 | 1.0 |
| 2004 | 400 | 2725 | 3.0 | 0.9 |
| 2004 | 600 | 2892 | 1.9 | 1.1 |
| 2004 | 800 | 3617 | 0.9 | 1.2 |
| 2004 | 1000 | 5096 | −0.1 | 1.3 |
| 2004 | 1200 | 6926 | −1.1 | 1.5 |
| 2004 | 1400 | 8141 | −2.2 | 1.6 |
| 2004 | 1600 | 7132 | −3.5 | 1.9 |
| 2004 | 1800 | 5762 | −4.9 | 2.2 |
| 2004 | 2000 | 4545 | −6.0 | 2.3 |
| 2004 | 2200 | 3426 | −6.8 | 2.1 |
| 2004 | 2400 | 2588 | −7.2 | 1.8 |
| 2004 | 2600 | 1797 | −8.1 | 1.8 |
| 2004 | 2800 | 1237 | −9.1 | 2.0 |
| 2004 | 3000 | 750 | −10.0 | 2.3 |
| 2004 | 3200 | 493 | −10.5 | 2.1 |
| 2004 | 3400 | 350 | −11.1 | 2.2 |
| 2004 | 3600 | 187 | −11.7 | 2.1 |
| 2004 | 3800 | 151 | −11.9 | 1.9 |
| 2004 | 4000 | 103 | −13.2 | 2.6 |
| 2004 | 4200 | 79 | −13.4 | 2.4 |
| 2004 | 4400 | 46 | −14.8 | 2.2 |
| 2004 | 4600 | 28 | −16.2 | 2.6 |
| 2004 | 4800 | 19 | −17.5 | 3.0 |
| 2004 | 5000 | 8 | −19.6 | 2.7 |
| 2006 | 200 | 1417 | 2.3 | 1.0 |
| 2006 | 400 | 2720 | 1.2 | 1.0 |
| 2006 | 600 | 2890 | 0.1 | 1.2 |
| 2006 | 800 | 3594 | −1.0 | 1.3 |
| 2006 | 1000 | 5090 | −2.0 | 1.4 |
| 2006 | 1200 | 6935 | −3.0 | 1.6 |
| 2006 | 1400 | 8116 | −4.1 | 1.8 |
| 2006 | 1600 | 7074 | −5.3 | 2.1 |
| 2006 | 1800 | 5605 | −6.5 | 2.3 |
| 2006 | 2000 | 4419 | −7.3 | 2.2 |
| 2006 | 2200 | 3376 | −7.9 | 1.9 |
| 2006 | 2400 | 2569 | −8.4 | 1.7 |
| 2006 | 2600 | 1791 | −9.2 | 1.7 |
| 2006 | 2800 | 1238 | −10.1 | 2.1 |
| 2006 | 3000 | 739 | −10.8 | 2.5 |
| 2006 | 3200 | 489 | −11.1 | 2.1 |

**Table A3.** *Cont.*

| Year | Elevation Band (m) | Area (km²) | Mean Annual Temp (°C) | Standard Deviation |
|------|------|------|------|------|
| 2006 | 3400 | 345 | −11.5 | 2.0 |
| 2006 | 3600 | 185 | −11.8 | 1.4 |
| 2006 | 3800 | 150 | −12.4 | 1.5 |
| 2006 | 4000 | 103 | −13.2 | 1.7 |
| 2006 | 4200 | 78 | −13.5 | 1.7 |
| 2006 | 4400 | 46 | −13.6 | 1.2 |
| 2006 | 4600 | 28 | −13.9 | 1.2 |
| 2006 | 4800 | 18 | −14.7 | 1.0 |
| 2006 | 5000 | 8 | −13.9 | 0.6 |
| 2008 | 200 | 1402 | 2.4 | 0.9 |
| 2008 | 400 | 2707 | 1.4 | 1.1 |
| 2008 | 600 | 2875 | 0.3 | 1.3 |
| 2008 | 800 | 3571 | −0.8 | 1.4 |
| 2008 | 1000 | 5098 | −1.7 | 1.6 |
| 2008 | 1200 | 6949 | −2.6 | 1.9 |
| 2008 | 1400 | 8103 | −3.6 | 2.0 |
| 2008 | 1600 | 7035 | −4.8 | 2.2 |
| 2008 | 1800 | 5511 | −6.0 | 2.3 |
| 2008 | 2000 | 4365 | −7.1 | 2.4 |
| 2008 | 2200 | 3380 | −7.9 | 2.4 |
| 2008 | 2400 | 2583 | −8.3 | 2.0 |
| 2008 | 2600 | 1804 | −8.9 | 1.8 |
| 2008 | 2800 | 1240 | −9.6 | 2.0 |
| 2008 | 3000 | 738 | −10.3 | 2.4 |
| 2008 | 3200 | 488 | −10.5 | 2.1 |
| 2008 | 3400 | 347 | −10.9 | 1.9 |
| 2008 | 3600 | 183 | −11.2 | 1.4 |
| 2008 | 3800 | 151 | −11.9 | 1.5 |
| 2008 | 4000 | 103 | −12.6 | 1.8 |
| 2008 | 4200 | 78 | −13.0 | 1.7 |
| 2008 | 4400 | 46 | −12.7 | 1.1 |
| 2008 | 4600 | 28 | −13.1 | 1.1 |
| 2008 | 4800 | 19 | −14.0 | 1.3 |
| 2008 | 5000 | 8 | −13.0 | 0.5 |
| 2010 | 200 | 1369 | 2.8 | 0.9 |
| 2010 | 400 | 2685 | 1.7 | 1.1 |
| 2010 | 600 | 2867 | 0.6 | 1.4 |
| 2010 | 800 | 3566 | −0.6 | 1.5 |
| 2010 | 1000 | 5084 | −1.5 | 1.7 |
| 2010 | 1200 | 6932 | −2.5 | 2.0 |
| 2010 | 1400 | 8097 | −3.4 | 2.3 |
| 2010 | 1600 | 6984 | −4.6 | 2.5 |
| 2010 | 1800 | 5429 | −5.8 | 2.4 |
| 2010 | 2000 | 4300 | −6.8 | 2.3 |
| 2010 | 2200 | 3350 | −7.7 | 2.2 |
| 2010 | 2400 | 2578 | −8.2 | 1.9 |
| 2010 | 2600 | 1802 | −8.9 | 1.7 |
| 2010 | 2800 | 1246 | −9.7 | 1.8 |
| 2010 | 3000 | 745 | −10.3 | 2.1 |
| 2010 | 3200 | 491 | −10.7 | 2.0 |
| 2010 | 3400 | 348 | −11.3 | 2.2 |
| 2010 | 3600 | 184 | −11.8 | 1.8 |
| 2010 | 3800 | 151 | −12.3 | 2.1 |
| 2010 | 4000 | 103 | −13.3 | 2.5 |
| 2010 | 4200 | 79 | −13.4 | 2.3 |
| 2010 | 4400 | 47 | −13.5 | 1.0 |

**Table A3.** *Cont.*

| Year | Elevation Band (m) | Area (km²) | Mean Annual Temp (°C) | Standard Deviation |
|------|------|------|------|------|
| 2010 | 4600 | 27 | −13.8 | 1.0 |
| 2010 | 4800 | 19 | −14.8 | 1.0 |
| 2010 | 5000 | 8 | −14.0 | 1.2 |
| 2012 | 200 | 1324 | 3.0 | 0.9 |
| 2012 | 400 | 2656 | 1.9 | 1.2 |
| 2012 | 600 | 2842 | 0.7 | 1.5 |
| 2012 | 800 | 3512 | −0.4 | 1.7 |
| 2012 | 1000 | 5021 | −1.3 | 1.9 |
| 2012 | 1200 | 6926 | −2.1 | 2.2 |
| 2012 | 1400 | 8066 | −3.0 | 2.5 |
| 2012 | 1600 | 6975 | −4.2 | 2.8 |
| 2012 | 1800 | 5467 | −5.4 | 2.7 |
| 2012 | 2000 | 4341 | −6.5 | 2.7 |
| 2012 | 2200 | 3389 | −7.4 | 2.7 |
| 2012 | 2400 | 2596 | −8.0 | 2.3 |
| 2012 | 2600 | 1808 | −8.7 | 2.0 |
| 2012 | 2800 | 1243 | −9.4 | 2.0 |
| 2012 | 3000 | 740 | −10.0 | 2.2 |
| 2012 | 3200 | 488 | −10.6 | 2.2 |
| 2012 | 3400 | 346 | −11.0 | 2.4 |
| 2012 | 3600 | 184 | −11.5 | 2.1 |
| 2012 | 3800 | 153 | −12.4 | 2.1 |
| 2012 | 4000 | 103 | −13.5 | 2.1 |
| 2012 | 4200 | 78 | −13.8 | 2.1 |
| 2012 | 4400 | 46 | −13.3 | 1.1 |
| 2012 | 4600 | 27 | −13.8 | 1.2 |
| 2012 | 4800 | 18 | −14.9 | 1.2 |
| 2012 | 5000 | 8 | −13.7 | 1.1 |
| 2014 | 200 | 1285 | 5.1 | 0.9 |
| 2014 | 400 | 2638 | 4.1 | 1.2 |
| 2014 | 600 | 2832 | 2.9 | 1.4 |
| 2014 | 800 | 3436 | 1.8 | 1.5 |
| 2014 | 1000 | 4906 | 1.0 | 1.6 |
| 2014 | 1200 | 6785 | 0.2 | 2.0 |
| 2014 | 1400 | 8039 | −0.6 | 2.4 |
| 2014 | 1600 | 7044 | −1.6 | 2.7 |
| 2014 | 1800 | 5535 | −2.8 | 2.7 |
| 2014 | 2000 | 4376 | −3.8 | 2.7 |
| 2014 | 2200 | 3390 | −4.5 | 2.7 |
| 2014 | 2400 | 2596 | −5.1 | 2.3 |
| 2014 | 2600 | 1809 | −5.6 | 2.0 |
| 2014 | 2800 | 1239 | −6.3 | 2.0 |
| 2014 | 3000 | 743 | −6.8 | 2.1 |
| 2014 | 3200 | 486 | −7.3 | 2.0 |
| 2014 | 3400 | 343 | −7.9 | 2.2 |
| 2014 | 3600 | 180 | −8.2 | 1.8 |
| 2014 | 3800 | 152 | −8.9 | 2.1 |
| 2014 | 4000 | 103 | −9.8 | 2.1 |
| 2014 | 4200 | 77 | −9.9 | 1.6 |
| 2014 | 4400 | 44 | −10.0 | 1.4 |
| 2014 | 4600 | 26 | −10.1 | 1.4 |
| 2014 | 4800 | 18 | −10.8 | 1.3 |
| 2014 | 5000 | 6 | −9.6 | 1.4 |
| 2016 | 200 | 1254 | 3.4 | 1.0 |
| 2016 | 400 | 2615 | 2.2 | 1.3 |
| 2016 | 600 | 2788 | 1.0 | 1.5 |

**Table A3.** *Cont.*

| Year | Elevation Band (m) | Area (km²) | Mean Annual Temp (°C) | Standard Deviation |
|------|------|------|------|------|
| 2016 | 800 | 3371 | −0.2 | 1.6 |
| 2016 | 1000 | 4786 | −1.0 | 1.7 |
| 2016 | 1200 | 6657 | −1.9 | 2.0 |
| 2016 | 1400 | 7925 | −2.7 | 2.3 |
| 2016 | 1600 | 7103 | −3.8 | 2.6 |
| 2016 | 1800 | 5690 | −5.0 | 2.6 |
| 2016 | 2000 | 4513 | −5.8 | 2.5 |
| 2016 | 2200 | 3475 | −6.4 | 2.4 |
| 2016 | 2400 | 2582 | −6.9 | 2.1 |
| 2016 | 2600 | 1799 | −7.6 | 2.0 |
| 2016 | 2800 | 1232 | −8.4 | 2.1 |
| 2016 | 3000 | 735 | −9.1 | 2.4 |
| 2016 | 3200 | 477 | −9.3 | 2.3 |
| 2016 | 3400 | 335 | −9.9 | 2.5 |
| 2016 | 3600 | 178 | −10.3 | 2.2 |
| 2016 | 3800 | 147 | −10.8 | 2.3 |
| 2016 | 4000 | 102 | −11.7 | 2.7 |
| 2016 | 4200 | 75 | −11.6 | 1.9 |
| 2016 | 4400 | 43 | −12.8 | 0.9 |
| 2016 | 4600 | 26 | −13.2 | 1.2 |
| 2016 | 4800 | 18 | −13.9 | 1.4 |
| 2016 | 5000 | 5 | −15.2 | 0.5 |
| 2018 | 200 | 1210 | 5.2 | 0.8 |
| 2018 | 400 | 2590 | 4.2 | 1.0 |
| 2018 | 600 | 2756 | 3.1 | 1.2 |
| 2018 | 800 | 3336 | 2.0 | 1.3 |
| 2018 | 1000 | 4735 | 1.3 | 1.4 |
| 2018 | 1200 | 6585 | 0.5 | 1.8 |
| 2018 | 1400 | 7885 | −0.3 | 2.2 |
| 2018 | 1600 | 7133 | −1.4 | 2.6 |
| 2018 | 1800 | 5796 | −2.5 | 2.7 |
| 2018 | 2000 | 4544 | −3.3 | 2.6 |
| 2018 | 2200 | 3499 | −3.8 | 2.4 |
| 2018 | 2400 | 2586 | −4.1 | 1.9 |
| 2018 | 2600 | 1801 | −4.7 | 1.7 |
| 2018 | 2800 | 1236 | −5.4 | 1.9 |
| 2018 | 3000 | 735 | −6.0 | 2.2 |
| 2018 | 3200 | 474 | −6.2 | 1.9 |
| 2018 | 3400 | 334 | −6.8 | 2.3 |
| 2018 | 3600 | 180 | −7.1 | 1.8 |
| 2018 | 3800 | 147 | −7.6 | 2.1 |
| 2018 | 4000 | 99 | −8.4 | 2.5 |
| 2018 | 4200 | 72 | −8.3 | 1.8 |
| 2018 | 4400 | 43 | −9.1 | 1.0 |
| 2018 | 4600 | 26 | −9.3 | 1.0 |
| 2018 | 4800 | 19 | −9.9 | 0.8 |
| 2018 | 5000 | 5 | −10.2 | 0.5 |
| 2020 | 200 | 1079 | −0.1 | 1.2 |
| 2020 | 400 | 2532 | −1.3 | 1.8 |
| 2020 | 600 | 2689 | −2.6 | 2.2 |
| 2020 | 800 | 3217 | −3.8 | 2.6 |
| 2020 | 1000 | 4521 | −4.4 | 2.8 |
| 2020 | 1200 | 6324 | −5.1 | 3.0 |
| 2020 | 1400 | 7666 | −5.9 | 3.3 |
| 2020 | 1600 | 7053 | −7.5 | 3.9 |

**Table A3.** *Cont.*

| Year | Elevation Band (m) | Area (km²) | Mean Annual Temp (°C) | Standard Deviation |
|---|---|---|---|---|
| 2020 | 1800 | 5868 | −9.4 | 4.0 |
| 2020 | 2000 | 4685 | −10.9 | 4.0 |
| 2020 | 2200 | 3517 | −11.8 | 3.9 |
| 2020 | 2400 | 2571 | −12.1 | 3.8 |
| 2020 | 2600 | 1797 | −12.6 | 3.7 |
| 2020 | 2800 | 1229 | −13.2 | 4.2 |
| 2020 | 3000 | 734 | −13.9 | 4.7 |
| 2020 | 3200 | 468 | −14.2 | 5.1 |
| 2020 | 3400 | 323 | −14.8 | 5.4 |
| 2020 | 3600 | 173 | −14.4 | 4.8 |
| 2020 | 3800 | 142 | −13.0 | 3.8 |
| 2020 | 4000 | 99 | −14.0 | 4.6 |
| 2020 | 4200 | 70 | −13.4 | 3.6 |
| 2020 | 4400 | 42 | −14.8 | 2.8 |
| 2020 | 4600 | 27 | −15.5 | 3.7 |
| 2020 | 4800 | 19 | −16.2 | 3.9 |
| 2020 | 5000 | 6 | −20.0 | 0.7 |

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
