# Peer review of "Changes over the Last 35 Years in Alaska’s Glaciated Landscape: A Novel Deep Learning Approach to Mapping Glaciers at Fine Temporal Granularity"

_remotesensing, doi:10.3390/rs14184582_

Round 1

Reviewer 1 Report

General Comments:

This paper developed a novel deep learning method to automate the extraction of debris-free and debris-covered glaciers. Using the method, the author mapped the glacier coverage in Alaska over three decades with a two-year time interval. The dataset could provide more insights into the glacier changes in Alaska, and the method has the potential to be applied to broader regions. With that being said, some technical details need to be clarified before getting published. Below I will divide my comments into major and minor, and I hope they can explain my concerns.

Major Comments:

1. Class label generation

Training data is crucial for supervised deep learning applications. However, some technical details about class label generation need to be clarified. Following are several questions about label generation.

(1)  There are 18 biannual image composites. Do all of them have a corresponding label image? If not, which of the 18 biannual image composites have label images? In other words, how to pair the input images and target output labels?

(2)  From what I understand, STEM provides the label for debris-free ice, and RGI provides the label for debris-covered and debris-free ice. Therefore, the difference between STEM and RGI is the label for debris-covered ice. What puzzles me is the glacier probability layer's role in generating the class label.

(3)  How is the glacier probability leveraged? Are the pixels with values larger than one considered glacier and the rest considered non-glacier?

(4)  Are class labels temporally varying? Or there is one class label without a timing.

2. Data Split

Following the deep learning convention, there should be three sets: training, validation, and test. Validation is for choosing hyperparameters such as stride size and partition sizes. After the hyperparameters are selected, and the deep learning model is well trained, the model should be applied to the third set, namely the test set to quantify the accuracy of the deep learning model. All the error analyses should be conducted in the test set since high accuracy is expected in the training and validation sets.

Minor Comments:

Line 137: It would be better if the author could provide more information about spectrally homogenous time series.

Line 271: How to deal with the overlaps?

Line 298: PSPNETPSPNet

Line 299: input imagery decoder  input imagery

Line 300: It would be better to have a diagram of the network

Line 301: Explain more about the difference between northern and southern; a figure would be better.

Line 325: Are there two GlacierCoverNets (same structure, different sets of model parameters), one for mapping debris-free ice and the other for debris-covered ice?

Line 401: Are these two examples from the training or test set?

Line 402: Since the RGI is used to generate the class labels and GlacierCoverNet is the supervised deep learning method, it would be expected that the output of GlacierCoverNet to be close to RGI but not superior to it. So, what may lead to the higher fidelity of GlacierCoverNet’s output?

Table 3: Are the values here representing the validation set? If I understand correctly, the area in the first and second rows should be the area of both true positive and false positive. Please double-check the values, the area of false negative and true positive should equal the area of RGI.

Table 3: Glacier areas are temporally varying. Which year is the RGI representing? The difference between GlacierCoverNet outputs and RGI includes both method’s uncertainties and area variations.

Line 523: It should be the percentage of the true positive.

Reviewer 2 Report

This documents an excellent initiative using deep learning methodology with a time sequence of glacial imagery and addressing the growing issue of debris cover. My comments mostly address the figures and tables - where in a number of cases I can't see where the data results are tabulated - maybe I'm missing something !? In the case of overly precise graticule values, this is an all too common default feature in GIS software that should be easily over-ridden.

Reviewer 3 Report

This study produced a 35-year dataset of debris-free glaciers and supraglacial debris distributions in Alaska by developing a new deep learning-based model, GlacierCoverNet, which has the following advantages over previous study (1) high temporal resolution of 2 years; (2) the production process is highly automated with few manual edits, yet the accuracy of the results is comparable to manual or semi-automated results; (3) the model producing this dataset are expected to be migrated to similar glacial landscapes or even globally, enabling large-scale automatic extraction.

Improving suggestions:

(1) 1. Introduction: Important interactions between debris-free glaciers and supraglacial debris are not elucidated, which is an important reason to distinguish between them in this study dataset.

Suggestion: Add an overview of research progress and relevant literature on debris-free glaciers and supraglacial debris interactions.

(2) 3.2.1 Overall glacier covered area: GlacierCoverNet debris-free glaciers training data were generated by the STEM model, and supraglacial debris training data were obtained by subtracting debris-free glaciers generated by the STEM model from RGI6.0 glacier data. In other words, when making training data, RGI6.0 has been used by default for all glaciers, and the training results are then compared with RGI6.0, which is equivalent to comparing the training results with the training data, so the conclusions drawn are not convincing.

Suggestion: remove this part.

(3) 3.2.2 Supraglacial debris error analysis: The results of deep learning depend on the quality of the training data, and the training data in this study are mainly generated by the STEM model, but this paper does not compare the results of GlacierCoverNet and STEM, so the reader has no way to know how much the STEM method contributes to the GlacierCoverNet method, and whether adding the GlacierCoverNet method has an improvement over using only the STEM method?

Suggestion: add a comparison of GlacierCoverNet and STEM results to the result comparison to see if the secondary use of STEM (adding deep learning methods) has improved compared to using only STEM methods.

Figures

(1) 1.1 Study area: Figure 1 does not show where Alaska is in the world.

Suggestion: Add a thumbnail of the world map to show the geographical location of Alaska in the world map.

(2) 2. Materials and Methods: Figure 2 GlacierCoverNet Training part of the connection line too many unnecessary twists and turns, not very beautiful.

Suggestion: Revise the connection lines in this section to make it simple and beautiful.

(3) 2.4.1 Encoder-Decoder structure: Only the text of GlacierCoverNet model is introduced, but there is no schematic diagram of the model structure.

Suggestion: Add the structure diagram of GlacierCoverNet model.

Overall

The actual contribution of the GlacierCoverNet method to the accuracy of the extraction results cannot be estimated, and the migratory properties are shown to be not so good. This study did not compare the STEM method alone with the STEM combined with the GlacierCoverNet method, so it is not possible to measure the actual contribution of the GlacierCoverNet method to the extraction results and whether there is an improvement over the STEM method alone. In addition, the model training area is in the south, resulting in poor extraction results in the northern Brooks Range region (this part of the results was removed in the results analysis), which shows the poor migration of the model, which limits the application of the model in similar glacial landform regions and even globally.

Round 2

Reviewer 1 Report

Most of the comments have been addressed carefully, and I appreciate the well-documented answer prepared by the author. Given this, I recommend this paper for publication.

The only thing that I am concerned is about Point 18. The percentage that I mean is in Line 561: “In 2020, GlacierCoverNet identified an area of glacier cover that was 97% of the area identified in RGI 6.0”. The total number is 97% of RGI 6.0 but it is the sum of both true positive and false positive. And the false positive area is not identified by RGI 6.0. Also, the author uses this 97% as the final accuracy in the abstract, which I think is debatable. I think the percentage of true positive is more appropriate to represent the accuracy.

Minor comments:

Line 261: stemSTEM

Reviewer 3 Report

The authors have addressed all my comments, and I have no more comments and agree with the publication of this manuscript. 

Author Response

Please see the attachment (a response was required to proceed). 
